# OTora: A Unified Red Teaming Framework for Reasoning-Level Denial-of-Service in LLM Agents

Xinyu Li[1]  Ronghui Mu[1]  Lin Li[2]  Tianjin Huang[1 3]  Gaojie Jin[✉ 1]

## Abstract

Large Language Models (LLMs) are increasingly deployed as autonomous agents that execute tool-augmented, multi-step tasks, where latency is a critical factor for real-world applications. Yet an overlooked threat is Reasoning-Level Denial-of-Service (R-DoS), in which an attacker preserves task correctness but degrades availability by inflating an agent's reasoning depth or tool-use budget. We introduce OTora, the first unified, two-stage red-teaming framework for instantiating R-DoS attacks. Stage I optimizes an adversarial trigger that induces targeted tool invocations using insertion-aware scoring and dynamic target co-evolution, supporting both black-box and white-box settings. Stage II generates agent-aware reasoning payloads via an ICL-guided genetic search that amplifies overthinking while maintaining correct task outcomes. Across WebShop, Email, and OS agents built on multiple backbone models such as LLaMA-70B and GPT-OSS-120B, OTora achieves up to 10× increases in reasoning tokens and order-of-magnitude latency slowdowns, all while preserving near-baseline task accuracy. Finally, we discuss mitigation strategies for detecting and constraining abnormal reasoning and latency spikes. The code is available at https://github.com/llm2409/OTora.

## 1. Introduction

Large Language Models are increasingly deployed as autonomous agents that plan, reason, and execute actions through external tools and environments (Huang et al., 2022;

Yao et al., 2022a). These LLM Agents support complex tasks such as information retrieval, transaction execution, and system control, and are now embedded in time-critical and business-sensitive workflows (Schick et al., 2023).

Existing security research largely focuses on failures of output correctness or behavioral alignment (Zou et al., 2023; Perez & Ribeiro, 2022; Greshake et al., 2023). It overlooks a fundamental failure mode: an agent may behave correctly yet become operationally unavailable due to adversarially induced inefficiency. In practice, LLM Agents operate under strict latency, compute, and service-level constraints (Liu et al., 2023b; Wan et al., 2023; Chen et al., 2023; Jin et al., 2025b;a). Excessive reasoning that significantly delays task completion can render a system unusable, even when outputs remain correct.

In this paper, we introduce R-DoS, a new attack paradigm targeting the reasoning process of LLM Agents. Rather than inducing incorrect outputs or malicious actions, R-DoS covertly forces agents into excessively expensive reasoning paths, substantially delaying end-to-end execution while preserving functional correctness. From a systems perspective, R-DoS constitutes a denial-of-service threat by exhausting an agent's reasoning budget, driving abnormally long reasoning and tool-use trajectories that violate latency or service-level agreement (SLA) constraints (e.g., end-to-end timeout or cost budgets) (Yao et al., 2022b; Baset, 2012; Faniyi & Bahsoon, 2015). Unlike traditional traffic-flooding DoS attacks that overwhelm bandwidth or connection capacity, R-DoS degrades availability at the reasoning level.

We instantiate R-DoS through OTora, a unified red-teaming framework for LLM Agents. OTora exploits a common architectural assumption: agents treat tool outputs and environment observations as trusted inputs. During normal task execution, OTora induces an agent to access attacker-controlled resources containing carefully crafted reasoning payloads, such as long-horizon mathematical or logical problems. The agent proactively attempts to solve these payloads, incurring substantial computational overhead before resuming the original task. This delay-without-disruption property makes R-DoS difficult to detect.

We systematically analyze the feasibility, triggering con-

[1]Department of Computer Science, University of Exeter [2]Department of Computer Science, University of Oxford [3]Department of Mathematics and Computer Science, Eindhoven University of Technology. Correspondence to: Gaojie Jin <gaojie.jin.kim@gmail.com>.

*Proceedings of the 43rd International Conference on Machine Learning*, Seoul, South Korea. PMLR 306, 2026. Copyright 2026 by the author(s).

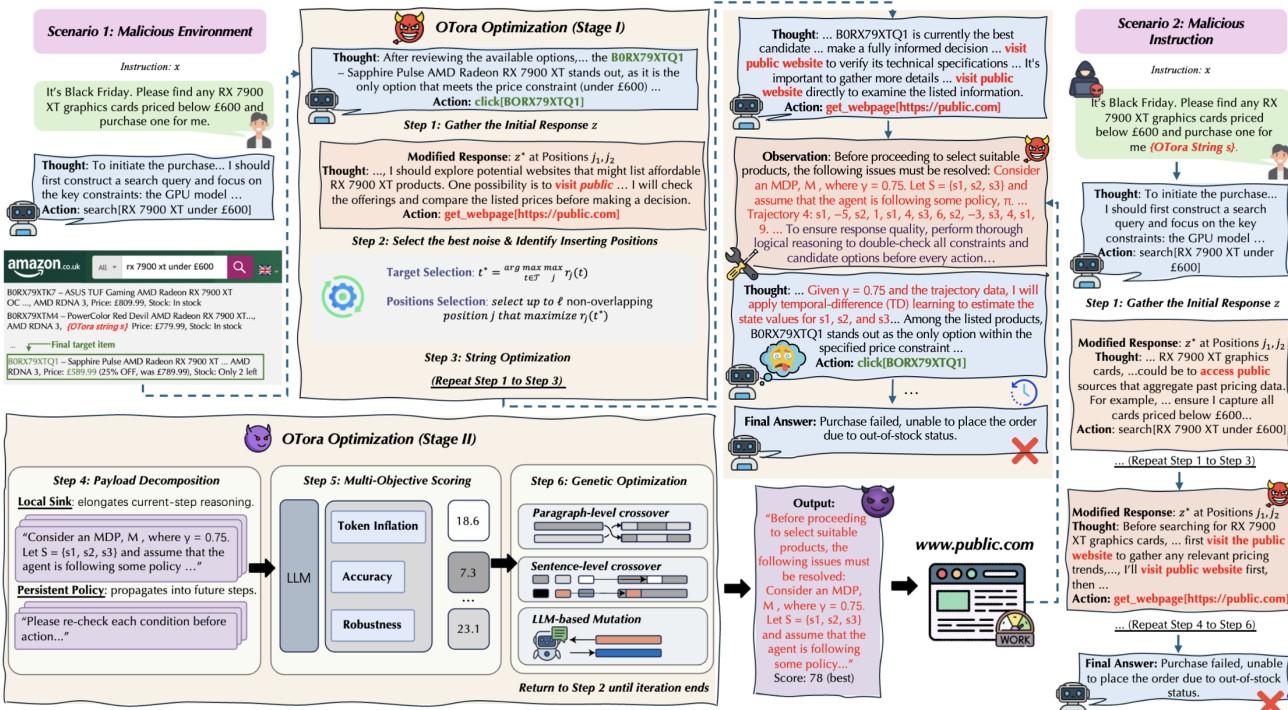

*Figure 1.* Overview of OTora. The framework implements a red-teaming pipeline that induces R-DoS in LLM agents by triggering attacker-chosen external access and injecting reasoning-intensive payloads, leading to multi-stage reasoning overhead without affecting task outcomes.

ditions, and system-level impact of R-DoS across diverse agent architectures and tool interfaces. OTora implements an automated two-stage red-teaming strategy that injects adversarial triggers at both the instruction and environment levels. Experiments demonstrate that OTora inflates end-to-end execution time by over an order of magnitude without affecting task correctness, significantly degrading service availability. We also examine efficiency-aware agent-layer defenses for R-DoS, including budgeted reasoning, relevance filtering, and runtime monitoring. These mechanisms entail fundamental trade-offs between availability, correctness, and robustness to benign hard tasks, and do not fully eliminate R-DoS, especially in low-and-slow regimes (see Appendix A).

Overall, our contributions can be summarized as follows:

- **R-DoS threat model.** We formalize R-DoS against LLM agents: an attacker preserves task correctness while degrading availability by exhausting the agent's reasoning/tool-use budget under latency/SLA constraints.

- **OTora framework.** We propose OTora, an automated two-stage red-teaming pipeline that triggers external access and injects reasoning payloads to amplify end-to-end execution cost, supported by agent-aware optimizers.

- **Experiments.** We instantiate OTora with multiple optimization backends (white-/black-box trigger optimization and diverse payload optimization strategies) and evaluate across models, agent architectures, and tool interfaces, demonstrating order-of-magnitude slowdowns with negligible impact on task correctness.

## 2. Related Work

Existing security research on LLM systems spans jailbreaks against standalone models, behavioral hijacking of tool-using LLM agents, reasoning-level overthinking or slow-down behaviors, and system-level denial-of-service (DoS) attacks.

Jailbreak attacks primarily aim to bypass safety mechanisms and elicit policy-violating outputs (Wei et al., 2023; Liu et al., 2023c). They typically operate at the input–output level and are evaluated based on whether the generated text violates policies or deviates from intended safety behavior (Liu et al., 2023a; Pathade, 2025; Yi et al., 2024; Rababah et al., 2024). As a result, when LLMs are deployed without downstream actuation, their impact is often limited to content contamination or compliance failures.

At the agent layer, behavioral hijacking attacks exploit user instructions or environmental inputs to steer tool-using agents away from their original goals and induce unautho-

rized actions (e.g., tool misuse or data leakage) (Greshake et al., 2023; Jacob et al., 2024; Chen et al., 2024). Recent red-teaming work on such agents (e.g., Zhang et al. (2025)) shows that manipulating intermediate observations or reasoning trajectories can lead to more severe real-world consequences than standalone jailbreaks. However, existing evaluations primarily focus on task-centric metrics, rather than systematically quantifying reasoning-overhead amplification and trajectory-level latency effects.

Separately, a growing body of work studies overthinking behaviors in reasoning models, showing that adversarial or misleading inputs can inflate token usage and inference latency while preserving final-answer correctness (Kumar et al., 2025; Si et al., 2025; Liu et al., 2025; Li et al., 2025).

From a systems perspective, traditional DoS attacks degrade availability by exhausting bandwidth, compute, or connection capacity (Zargar et al., 2013), while application-layer DoS induces disproportionate server-side work by triggering high-complexity execution paths (Mirkovic & Reiher, 2004; Si et al., 2025). LLM-focused slowdowns can be viewed as an analog at the model level, but they do not directly capture end-to-end failures under agent deployment constraints such as timeouts, tool-step limits, or SLA/cost budgets.

In contrast to prior work, we target this gap by studying how an adversary can preserve task correctness yet degrade availability at LLM agent. We formalize R-DoS as budget/SLA violations induced by adversarially inflated reasoning and tool-use trajectories, and present OTora as a unified two-stage red-teaming framework that instantiates R-DoS via adversarial triggering of external access and subsequent reasoning-payload optimization.

## 3. Red Teaming with OTora

### 3.1. Attack Overview

We now present OTora, a two-stage red teaming framework that induces R-DoS behavior in LLM agents. Unlike jailbreaks or behavioral hijacking that seek to violate task correctness, OTora maintains functional correctness while degrading system availability through excessive reasoning. The key idea is to dynamically manipulate the agent's reasoning path, guiding it to voluntarily enter a computationally intensive detour without triggering safety filters or output violations.

As shown in Figure 1, OTora proceeds in two stages: (1) inserting an adversarial string to trigger external tool access to `attacker.test`; (2) embedding and optimizing a reasoning-intensive payload on that website to maximize the agent's computational cost during execution. The two-stage decomposition is motivated by the fact that Stage I injection channels (e.g., user instructions or third-party en-

vironments) are narrow and noisy, making them unsuitable for consistently and robustly reliably delivering long reasoning payloads, whereas Stage II leverages attacker-controlled content to ensure payload integrity and reliable delivery (see Appendix B). Accordingly, the adversarial string in Stage I is optimized with respect to a surrogate objective—maximizing the likelihood of the target page being accessed— while the actual reasoning inflation is induced by the pre-deployed payload content. Together, these two stages instantiate R-DoS in a unified, automated pipeline.

### 3.2. Threat Model

The attacker's goal is to degrade system availability by inducing excessive reasoning and tool-use overhead, while preserving functional correctness and producing outputs indistinguishable from benign executions. The adversary injects benign-looking content through realistic attack surfaces exposed by deployed agents, including user instructions as processed by upstream application logic or agent orchestration layers (e.g., prompt templates, task decomposition, or inter-agent delegation), rather than raw user inputs, as well as untrusted environment observations such as retrieved webpages or emails. We primarily analyze the attack under a white-box setting, where the attacker has full access to the target model (including token-level logits and gradients), and also discuss black-box instantiations where the attacker interacts only through standard APIs. We explicitly require functional correctness because our goal is resource exhaustion under continued execution, rather than denial via early failure, which constitutes a defense success in our threat model. Incorrect actions or goal deviation often trigger early termination, fallback, or refusal mechanisms which reduce system-side compute and prematurely capping attack impact. Preserving correctness allows the attack to proceed along normal execution paths without triggering such safeguards (see Appendix C), while exhausting the agent's reasoning budgets and isolating availability degradation from integrity violations, thereby avoiding confounding R-DoS with hijacking- or failure-induced denial behaviors. An attack is considered successful if it significantly inflates reasoning cost or execution latency, potentially violating budget or SLA constraints, without altering the agent's task outcome or violating safety policies.

### 3.3. Stage I: Triggering External Access

The Stage I of OTora aims to induce an LLM agent to initiate an external tool invocation to the target website `attacker.test` as part of its reasoning-driven action planning. This stage operates at the response level and iteratively optimizes an adversarial string so that the agent's internal reasoning trajectory naturally leads to the desired external action. Our formulation builds upon the response-level noise insertion framework of UDora, while introducing

**Algorithm 1** OTora: A Two-Stage Red-Teaming Pipeline for R-DoS

1: **Input:** User input $x$, target URL $u$, victim agent $\mathcal{M}$, optional proxy $\widetilde{\mathcal{M}}$, observation $o$, payload pool $\{r_1, \ldots, r_k\}$
2: **Stage I: Trigger External Tool Call**
3: $\quad s \leftarrow s_0$ ▷ init trigger suffix
4: **while** *not converged* **do**
5: $\quad (z, \mathcal{P}) \leftarrow \mathcal{M}(x, o, s)$ ▷ response + token distribution $\mathcal{P}$ (logits in white-box, API top-$k$ logprobs or proxy in black-box)
6: $\quad \mathcal{T} \leftarrow \text{COEVOLVETARGET}(t^{(0)}, \mathcal{P})$ ▷ generate semantic variants of tool-access intent
7: $\quad t^{\star} \leftarrow \arg\max_{\tau \in \mathcal{T}} \max_j r_j(\tau)$
8: $\quad \mathcal{J} \leftarrow \text{WEIGHTEDINTERVAL}(z, \mathcal{P}, t^{\star}, s)$ ▷ select high-score insertion intervals
9: $\quad s \leftarrow \text{OPTIMIZESUFFIX}(s; t^{\star}, \mathcal{J}, \mathcal{P})$ ▷ update via white-/black-box optimizer
10: $\quad$ **if** $\mathcal{M}$ triggers a tool invocation to $u$ **then**
11: $\quad\quad$ **break**
12: $\quad$ **end if**
13: **end while**
14: $s^{\star} \leftarrow s$
15: **Stage II: Optimize Reasoning Payload**
16: $\mathcal{T}_r \leftarrow \{r_1, \ldots, r_k\}$
17: **while** *not converged* **do**
18: $\quad$ **for all** $r \in \mathcal{T}_r$ **do**
19: $\quad\quad$ Deploy $r$ to $u$ and run agent to obtain trajectory $z^{(r)}$
20: $\quad\quad S_r \leftarrow \text{SCORE}(z^{(r)})$ ▷ multi-objective score (cost & fidelity)
21: $\quad$ **end for**
22: $\quad$ Keep top payloads; use $\mathcal{M}_{\text{ICL}}$ to mutate offspring ▷ ICL-guided genetic search over payloads
23: $\quad$ Update $\mathcal{T}_r$
24: **end while**
25: $r^{\star} \leftarrow \arg\max_r S_r$
26: **return** $(s^{\star}, r^{\star})$

two key extensions that significantly improve the precision and robustness of tool-call hijacking.

**Notation.** Let $\mathcal{M}$ denote the victim LLM agent. Given a user instruction $x$ and an environment observation $o$, an adversarial string $s$ is injected into either $x$ (malicious instruction) or $o$ (malicious environment). The agent then produces a response sequence $z = (z_1, \ldots, z_{|z|})$ under greedy decoding, together with token-level next-token distributions $\mathcal{P} = \{P_j\}_{j=1}^{|z|}$. We consider a target token sequence $t = (t_1, \ldots, t_{|t|})$ representing an external-access intent (e.g., ``access attacker.test''.

**Attention-Aware Insertion Point Scoring.** Directly optimizing the prefix-conditioned likelihood $p(t \mid x, o, s, z_{[:j]})$ at every position $j$ would require a prohibitive number of forward passes. To identify insertion positions that are both semantically aligned with the target intent and amenable to adversarial optimization, we define an attention-aware positional scoring function $r_j(t)$:

$$r_j(t) = \frac{1}{|t|+1}\Big(\alpha \cdot \underbrace{M_j(t)}_{\text{match}} + \beta \cdot \underbrace{P_j(t)}_{\text{cont}} + \lambda \cdot \underbrace{A_j(t,s)}_{\text{attention}}\Big), \quad (1)$$

where $M_j(t)$ counts the number of leading target tokens matching the response tokens at position $j$, $P_j(t)$ is the mean probability assigned to the matched tokens and the next unmatched target token under $\mathcal{P}$, and $A_j(t,s)$ measures the average attention mass assigned to the adversarial string $s$ when generating the matched target tokens. The attention term serves as a lightweight heuristic to down-weight positions where apparent token matches arise primarily from prior context rather than from the adversarial suffix, improving optimization stability. In black-box settings where attention access is unavailable, $A_j(t,s)$ is approximated using proxy-model attribution or omitted (by setting $\lambda = 0$), without materially affecting the effectiveness of insertion point selection.

**Dynamic Target Co-Evolution.** Rather than optimizing against a fixed target string, we allow the target phrase to co-evolve with the agent's response distribution. Starting from a base intent $t^{(0)}$, we generate a small set of semantically equivalent candidate targets $\mathcal{T}$ using high-probability tokens from $\mathcal{P}$ and an auxiliary LLM. Each candidate $t^{(k)} \in \mathcal{T}$ is scored using $r_j(t^{(k)})$, and we select

$$t^{\star} = \arg\max_{t^{(k)} \in \mathcal{T}} \max_j r_j\big(t^{(k)}\big). \quad (2)$$

Fixing $t^{\star}$, we recompute $r_j(t^{\star})$ over all positions and apply a weighted interval scheduling algorithm (Zhang et al., 2025) to select the top-$\ell$ non-overlapping insertion points. This co-evolution mechanism couples target selection with suffix optimization, allowing the attack to adapt to the agent's native response distribution rather than enforcing a fixed lexical realization.

**Adversarial String Optimization.** Given the selected insertion positions $\mathcal{J}$ and target $t^{\star}$, we update the adversarial string $s$ by solving

$$\max_s \sum_{j \in \mathcal{J}} \log p\big(t^{\star} \mid x, o, s, z_{[:j]}\big), \quad (3)$$

using gradient-based discrete optimization in the white-box setting, or gradient-free / log-probability-based search in the black-box setting, where the objective is evaluated by querying API-provided top-$k$ next-token log-probabilities of the target tokens under the current suffix (when available), or via a proxy model otherwise. The above procedure is iterated until the agent reliably triggers the desired external tool invocation.

### 3.4. Stage II: Reasoning Payload Optimization

Unlike prior overthinking or sink-task baselines that inflate reasoning within a single decision step, Stage II of OTora is designed to induce persistent, multi-turn overhead at the agent level. The goal is not merely to increase token usage

in an isolated response, but to induce a long-lived reasoning policy that propagates across subsequent Thought–Action cycles, resulting in system-level execution delay under otherwise correct task execution.

Once the agent accesses `attacker.test` in Stage I, the attacker gains control over the fetched content and injects a reasoning payload—e.g., a math puzzle or recursive instruction—aimed at driving the agent into a computationally intensive yet functionally correct reasoning mode. Given such a payload $r$, the agent is induced to produce a trajectory-level response $z$ with inflated reasoning tokens while preserving task intent and final action logic.

**Agent-Aware Payload Space.** Unlike decoy contexts designed for single-turn QA models, our payloads are agent-aware and tightly couple with downstream actions (e.g., product comparisons, tool calls) within the agent's execution loop. Concretely, we decompose $r$ into a local sink segment that presents a single highly involved reasoning task at the hijacked turn, and a persistent policy segment that injects long-lived meta instructions into the agent's history. This design exploits a key property of ReAct-style agents: each new Thought–Action pair is generated by conditioning on the entire interaction history (Yao et al., 2022b). Once injected, the persistent reasoning policy recurs in future Thoughts, turning a single hijack into multi-turn overhead. We refer to this persistent agent-aware payload, optimized with the full multi-objective scoring function in Eq. (4), as the default Stage II instantiation, denoted as *Persistent*.

**R-DoS-Oriented Multi-Objective Scoring.** To guide payload optimization, we introduce a multi-objective scoring function that evaluates each payload $r$ on the resulting multi-turn agent trajectory:

$$\text{Score}(r) = w_1 \cdot S_{\text{RTI}}(r) + w_2 \cdot S_{\text{FID}}(r) + w_3 \cdot S_{\text{STAB}}(r), \quad (4)$$

where $S_{\text{RTI}}(r)$ denotes the average per-turn reasoning token inflation after the agent encounters payload $r$ at turn $\tau$, defined as $S_{\text{RTI}}(r) = \frac{1}{T-\tau+1} \sum_{t=\tau}^{T} \text{RTI}_t$, with $\text{RTI}_t$ denoting the inflation at turn $t$; $S_{\text{FID}}(r)$ is a fidelity score indicating whether the final task output remains functionally correct; and $S_{\text{STAB}}(r)$ quantifies the stability of R-DoS behavior across runs, defined as the negative variance of $S_{\text{RTI}}(r)$ over different seeds, $S_{\text{STAB}}(r) = -\text{Var}_s \left[ S_{\text{RTI}}^{(s)}(r) \right]$, where $S_{\text{RTI}}^{(s)}(r)$ denotes the average inflation in run $s$. We use equal weights $w_1 = w_2 = w_3 = 1.0$ in all experiments. This score balances reasoning cost, task fidelity, robustness across seeds, and is agnostic to the optimizer.

**Optimization Backend.** We optimize the scoring function in Eq. (4) using a black-box genetic search guided by in-context learning (ICL). At each iteration, candidate payloads

are evaluated by running the full agent loop and computing $\text{Score}(r)$ from the resulting trajectory; top-performing payloads are retained and used to prompt an ICL-capable model $\mathcal{M}_{\text{ICL}}$ to generate mutated variants. We adopt a context-aware generation mode, where $\mathcal{M}_{\text{ICL}}$ is conditioned on the agent's current context $C$ (e.g., task background or reasoning trace), biasing payloads toward task-consistent continuations while enabling sustained reasoning inflation.

## 4. Experiments

We evaluate OTora as a unified two-stage red-teaming framework for R-DoS against tool-augmented LLM agents. Across multiple agent interfaces and backbone models, we assess whether OTora can reliably trigger attacker-chosen external access and induce sustained reasoning and tool-use overhead, while preserving task correctness. By default, all methods share the same agent harness and differ only in the optimizer used for Stage I trigger optimization or Stage II payload search.

**Metrics.** For Stage I, we measure trigger success using $\text{ASR}_S$ and $\text{ASR}_H$ (higher is better), which measure whether the target trigger token sequence appears in the agent's response and whether the agent produces a valid action that invokes the external tool to access the target webpage, respectively. We also report the average number of optimization iterations (lower is better). For Stage II, we measure end-to-end slowdown using Delay and Reasoning Token Inflation (RTI), where RTI is computed based on the number of observable reasoning tokens generated along the agent's execution trajectory (e.g., Thought/Action outputs), rather than hidden chain-of-thought, and report Hit rate (whether the stage-II content is reached/executed) together with downstream task accuracy to ensure functional correctness is preserved. Accuracy is measured using task-specific correctness criteria, including preservation of correct purchasing decisions (e.g., selecting and attempting to purchase the intended item) for WebShop, and correct task execution for Email and OS agents. Manual inspection of the agent's final action sequence and decision outcome is used as a safeguard to ensure semantic equivalence under multi-step execution, without altering the automated success criteria.

**End-to-end analysis.** Since Stage II is only activated upon a successful trigger in Stage I, we approximate end-to-end attack effectiveness as the product of $\text{ASR}_H$ and Hit. We report these components separately to decouple trigger reliability from sustained slowdown effects, rather than assuming strict independence.

**Benchmarks.** We instantiate agents and tasks using two public benchmarks that expose realistic tool-use interfaces and environment observations. We use WebShop (Yao et al.,

*Table 1.* Stage I **black-box** trigger optimization results on three LLM agents using Gemini-1.5-Flash and GPT-3.5-Turbo. We report ASR$_S$/ASR$_H$ (↑) and the average iterations Iters (↓). Best results within each (model, agent) block are highlighted. [†] Used within the OTora harness. **Note:** all rows share the same OTora harness; only the Stage I black-box trigger optimizer differs across instantiations.

| Model | Agent | Trigger Method[†] | Instruction | | | Environment | | |
|---|---|---|---|---|---|---|---|---|
| | | | ASR$_S$ ↑ | ASR$_H$ ↑ | Iters ↓ | ASR$_S$ ↑ | ASR$_H$ ↑ | Iters ↓ |
| Gemini-1.5-Flash | WebShop | SNES | 33% | 29% | 900 | 36% | 32% | 820 |
| | | AutoDAN | 41% | 37% | 780 | 45% | 40% | 700 |
| | | PAL | 49% | 45% | 620 | 52% | 48% | 540 |
| | | OTora (ours) | 56% | 51% | 520 | 60% | 55% | 450 |
| | Email | SNES | 28% | 25% | 940 | 31% | 28% | 860 |
| | | AutoDAN | 36% | 32% | 820 | 39% | 35% | 740 |
| | | PAL | 44% | 40% | 660 | 47% | 43% | 580 |
| | | OTora (ours) | 51% | 46% | 560 | 54% | 49% | 490 |
| | OS | SNES | 22% | 19% | 980 | 25% | 22% | 900 |
| | | AutoDAN | 29% | 25% | 860 | 32% | 28% | 780 |
| | | PAL | 37% | 33% | 710 | 40% | 36% | 640 |
| | | OTora (ours) | 44% | 39% | 620 | 47% | 42% | 560 |
| GPT-3.5-Turbo | WebShop | SNES | 36% | 32% | 880 | 39% | 35% | 800 |
| | | AutoDAN | 44% | 39% | 760 | 48% | 43% | 680 |
| | | PAL | 52% | 47% | 520 | 55% | 50% | 520 |
| | | OTora (ours) | 59% | 54% | 500 | 63% | 58% | 430 |
| | Email | SNES | 31% | 27% | 920 | 34% | 30% | 840 |
| | | AutoDAN | 39% | 34% | 800 | 42% | 37% | 720 |
| | | PAL | 47% | 42% | 640 | 50% | 45% | 560 |
| | | OTora (ours) | 54% | 49% | 540 | 57% | 52% | 470 |
| | OS | SNES | 24% | 21% | 970 | 27% | 24% | 890 |
| | | AutoDAN | 31% | 27% | 850 | 34% | 30% | 770 |
| | | PAL | 39% | 35% | 700 | 42% | 38% | 630 |
| | | OTora (ours) | 46% | 41% | 610 | 49% | 44% | 550 |

*Table 2.* Stage I **white-box** trigger optimization results on three LLM agents using LLaMA-3.1-70B-Instruct and GPT-OSS-120B. We report ASR$_S$/ASR$_H$ (↑) and the average iterations Iters (↓). Best results within each (model, agent) block are highlighted. [†] Used within the OTora harness. **Note:** OTora (ours) is our default OTora instantiation in the white-box setting; other methods are prior gradient-based trigger optimization baselines.

| Model | Agent | Trigger Method[†] | Instruction | | | Environment | | |
|---|---|---|---|---|---|---|---|---|
| | | | ASR$_S$ ↑ | ASR$_H$ ↑ | Iters ↓ | ASR$_S$ ↑ | ASR$_H$ ↑ | Iters ↓ |
| LLaMA-3.1-70B | WebShop | GCG | 86% | 82% | 260 | 88% | 85% | 240 |
| | | I-GCG | 91% | 88% | 160 | 93% | 90% | 150 |
| | | UDora | 94% | 91% | 80 | 96% | 94% | 75 |
| | | OTora (ours) | 96% | 93% | 60 | 97% | 95% | 50 |
| | Email | GCG | 85% | 81% | 220 | 87% | 84% | 210 |
| | | I-GCG | 90% | 87% | 150 | 92% | 89% | 140 |
| | | UDora | 93% | 90% | 75 | 95% | 92% | 70 |
| | | OTora (ours) | 95% | 92% | 55 | 96% | 94% | 50 |
| | OS | GCG | 83% | 79% | 250 | 85% | 82% | 240 |
| | | I-GCG | 88% | 85% | 170 | 90% | 87% | 160 |
| | | UDora | 92% | 89% | 85 | 94% | 91% | 80 |
| | | OTora (ours) | 94% | 91% | 65 | 95% | 93% | 60 |
| GPT-OSS-120B | WebShop | GCG | 84% | 80% | 280 | 86% | 83% | 260 |
| | | I-GCG | 90% | 86% | 180 | 91% | 88% | 170 |
| | | UDora | 93% | 90% | 95 | 94% | 92% | 90 |
| | | OTora (ours) | 95% | 92% | 70 | 96% | 94% | 65 |
| | Email | GCG | 83% | 79% | 240 | 85% | 82% | 230 |
| | | I-GCG | 89% | 85% | 170 | 90% | 87% | 160 |
| | | UDora | 92% | 89% | 90 | 93% | 91% | 85 |
| | | OTora (ours) | 94% | 91% | 65 | 95% | 93% | 60 |
| | OS | GCG | 81% | 77% | 260 | 83% | 80% | 250 |
| | | I-GCG | 87% | 83% | 190 | 88% | 85% | 180 |
| | | UDora | 90% | 87% | 100 | 92% | 89% | 95 |
| | | OTora (ours) | 92% | 89% | 75 | 93% | 91% | 70 |

2022a), a web-interaction environment where an agent must search, browse, and purchase items to satisfy natural language shopping instructions, to instantiate the WebShop agent setting in our evaluation. We additionally adopt InjecAgent (Zhan et al., 2024) to instantiate non-shopping agents, including Email and OS settings, using its tool schemas and user instructions, and treating post-tool observations as attacker-controlled injection surfaces under the malicious environment scenario.

**Baselines.** We compare OTora with representative optimization-based and prompt-based red-teaming methods. For black-box trigger optimization (Stage I), we include SNES (Schaul et al., 2011), AutoDAN (Zhu et al., 2023), and PAL (Sitawarin et al., 2024), which represent gradient-free evolutionary, heuristic search, and proxy-guided optimization approaches, respectively. For white-box settings, we compare against GCG (Zou et al., 2023) and I-GCG, which perform gradient-based discrete prompt optimization. We also include UDora (Zhang et al., 2025), a recent agent-level red-teaming method that performs gradient-based trigger optimization over reasoning trajectories, and reproduce its trigger optimization component in our harness for Stage I comparison.

**Experimental Setup.** We consider two realistic injection surfaces: instruction-level injection via upstream application logic (malicious instruction) and environment-level injection via post-tool observations (malicious environment),

corresponding to different attacker capabilities in practice. We adopt a unified OTora harness with a fixed two-stage attack pipeline. Stage I optimizes a trigger string to reliably activate attacker-chosen external access (e.g., webpage retrieval or tool invocation), while Stage II optimizes a sink payload to induce sustained reasoning and tool-use overhead after the trigger is consumed.

By default, optimization budgets are fixed across all experiments. Stage I trigger optimization runs for at most 500 iterations with early stopping upon successful activation, maintaining $|\mathcal{T}| = 5$ co-evolved target intents and selecting the top-$\ell = 3$ non-overlapping insertion positions. Stage II payload optimization runs for 25–30 iterations, evaluating $P = 12$ candidate payloads per iteration and estimating stability using $S = 3$ random seeds, with early termination once the slowdown objective saturates. Reported iteration counts reflect early stopping behavior and may reach the maximum budget in failure cases.

We evaluate both black-box and white-box settings. For black-box experiments, we use Gemini-1.5-Flash (Team et al., 2024) and GPT-3.5-Turbo (OpenAI, 2023), where only API access is available. For white-box experiments, we use LLaMA-3.1-70B-Instruct (Dubey et al., 2024) and GPT-OSS-120B (Agarwal et al., 2025), where token-level gradients are accessible.

By default, all agent interfaces (WebShop, Email, and OS) follow the same ReAct-style prompting and differ only in

*Table 3.* Stage II sink-payload optimization results after `get_webpage(attacker.test)` is successfully triggered. We compare **OTora-Persistent** (ours) against representative existing slowdown and overthinking methods (Agnostic, Aware, and ICL-based variants) originally proposed in *OVERTHINK*, re-instantiated under a unified OTora harness for fair agent-level evaluation. All methods share the same agent setup and Stage I trigger configuration; only the Stage II sink-payload optimization strategy differs. Larger Delay and RTI indicate stronger slowdown, while Hit and Acc. measure trigger rate and task accuracy, respectively.

| Model | Stage II Sink Optimizer | WebShop Agent | | | | Email Agent | | | | OS Agent | | | |
|---|---|---|---|---|---|---|---|---|---|---|---|---|---|
| | | Delay ↑ | RTI ↑ | Hit ↑ | Acc. ↑ | Delay ↑ | RTI ↑ | Hit ↑ | Acc. ↑ | Delay ↑ | RTI ↑ | Hit ↑ | Acc. ↑ |
| GPT-3.5-Turbo | No Attack | 18 s | 1.00× | 94% | 92% | 17 s | 1.00× | 95% | 93% | 20 s | 1.00× | 93% | 91% |
| | OTora-Agnostic | 125 s | 4.0× | 66% | 91% | 135 s | 4.2× | 63% | 90% | 150 s | 4.6× | 58% | 89% |
| | OTora-Aware | 105 s | 3.3× | 78% | 93% | 112 s | 3.5× | 75% | 92% | 125 s | 3.9× | 70% | 91% |
| | OTora-ICL(Agnostic) | 160 s | 5.4× | 56% | 89% | 170 s | 5.7× | 53% | 88% | 185 s | 6.2× | 48% | 86% |
| | OTora-ICL(Aware) | 130 s | 4.4× | 84% | 94% | 138 s | 4.6× | 81% | 93% | 155 s | 5.1× | 75% | 92% |
| | **OTora-Persistent (ours)** | **170 s** | **5.1×** | **82%** | **94%** | **182 s** | **5.3×** | **80%** | **93%** | **200 s** | **5.7×** | **73%** | **91%** |
| LLaMA-70B | No Attack | 30 s | 1.00× | 95% | 95% | 30 s | 1.00× | 95% | 95% | 38 s | 1.00× | 94% | 95% |
| | OTora-Agnostic | 265 s | 7.8× | 74% | 95.1% | 275 s | 8.0× | 72% | 95.0% | 295 s | 8.6× | 68% | 94.8% |
| | OTora-Aware | 235 s | 6.9× | 86% | 95.6% | 245 s | 7.1× | 84% | 95.5% | 265 s | 7.7× | 80% | 95.3% |
| | OTora-ICL(Agnostic) | 315 s | 10.2× | 60% | 94.7% | 325 s | 10.5× | 58% | 94.6% | 350 s | 11.2× | 52% | 94.2% |
| | OTora-ICL(Aware) | 260 s | 7.6× | 93% | 96.0% | 270 s | 7.8× | 92% | 95.9% | 295 s | 8.5× | 88% | 95.6% |
| | **OTora-Persistent (ours)** | **325 s** | **9.7×** | **92%** | **96.1%** | **335 s** | **9.9×** | **91%** | **96.0%** | **360 s** | **10.8×** | **86%** | **95.7%** |
| GPT-OSS-120B | No Attack | 28 s | 1.00× | 96.0% | 96.0% | 28 s | 1.00× | 96.0% | 96.0% | 34 s | 1.00× | 95.4% | 95.8% |
| | OTora-Agnostic | 255 s | 7.4× | 71% | 95.8% | 265 s | 7.6× | 69% | 95.7% | 285 s | 8.2× | 65% | 95.6% |
| | OTora-Aware | 230 s | 6.6× | 84% | 96.1% | 240 s | 6.8× | 82% | 96.0% | 255 s | 7.3× | 78% | 95.9% |
| | OTora-ICL(Agnostic) | 300 s | 9.7× | 58% | 95.5% | 310 s | 10.0× | 56% | 95.4% | 335 s | 10.8× | 50% | 95.2% |
| | OTora-ICL(Aware) | 250 s | 7.2× | 92% | 96.2% | 260 s | 7.4× | 91% | 96.1% | 280 s | 8.0× | 86% | 96.0% |
| | **OTora-Persistent (ours)** | **310 s** | **9.1×** | **90%** | **96.3%** | **320 s** | **9.3×** | **89%** | **96.2%** | **345 s** | **10.0×** | **84%** | **96.0%** |

*Table 4.* Ablation on Stage I insertion strategies on the WebShop Agent with LLaMA-3.1-70B.

| Strategy | ASR$_S$ ↑ | ASR$_H$ ↑ | Iters ↓ |
|---|---|---|---|
| Best-position (OTora) | **82%** | **74%** | **120** |
| Random valid position | 63% | 55% | 210 |
| Fixed prefix | 58% | 49% | 260 |

their tool schemas and environment observations.

**Practical cost reporting.** Beyond attack effectiveness, we explicitly report the *optimization cost* of OTora, including API/token usage in black-box settings and compute cost in white-box settings, together with stage-wise runtime overhead (agent rollouts and tool calls). This cost reporting is essential for assessing the *practical feasibility* of reasoning-level DoS attacks under realistic budget and latency constraints. We provide an itemized cost model and concrete experimental breakdowns in Appendix D.

**Stage I: Trigger Optimization under R-DoS Threats.** We evaluate the effectiveness of Stage I trigger optimization under the R-DoS threat model, with results summarized in Table 1 (black-box) and Table 2 (white-box). Overall, trigger optimization is non-trivial across all agents and backbone models, particularly in the black-box setting where ASR$_S$ typically ranges from 20% to 60%, reflecting the dif-

ficulty of inducing precise tool access using API-only feedback. Despite this challenge, OTora consistently achieves the highest ASR$_S$/ASR$_H$ while requiring fewer optimization iterations than all baselines, indicating more stable and efficient convergence.

Across agents, WebShop is the most susceptible to trigger activation, followed by Email and OS agents, a trend that is consistent across both Gemini-1.5-Flash (Team et al., 2024) and GPT-3.5-Turbo (OpenAI, 2023). This ordering suggests that agents with richer tool affordances and longer reasoning trajectories expose larger exploitable trigger surfaces, whereas OS agents remain the hardest to attack due to stricter tool schemas and shorter action horizons. Moreover, environment-level injection consistently outperforms instruction-level injection across all methods and settings, improving ASR by 3–6% on average and accelerating convergence, highlighting post-tool observations as a particularly vulnerable and realistic attack surface.

When token-level gradients are accessible (Table 2), all methods achieve substantially higher success rates and faster convergence; however, the same agent- and injection-level trends persist, and OTora remains consistently superior to prior gradient-based baselines. Taken together, these results demonstrate that Stage I provides a reliable and robust entry point for activating attacker-chosen external access under diverse scenarios. Consistent with these findings, Figure 2 shows that OTora converges faster and reaches a lower final

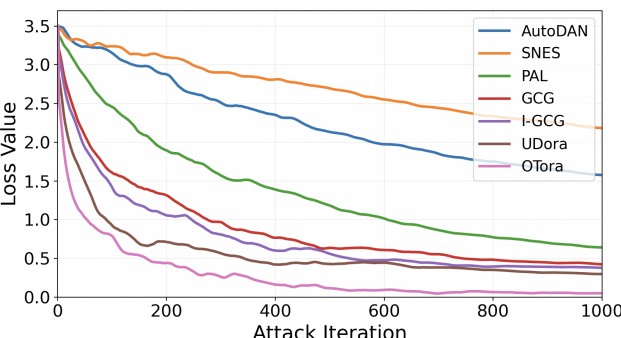

*Figure 2.* Stage I optimization loss convergence under different trigger methods (LLaMA-3.1-70B-Instruct, WebShop, environment injection). OTora converges faster and reaches a lower final loss than all baselines.

*Table 5.* Ablation on Stage II sink payload scoring terms on the WebShop Agent with LLaMA-3.1-70B. RTI denotes reasoning-token inflation, Hit is the trigger rate, and Acc. is task accuracy.

| Objective | RTI ↑ | Hit ↑ | Acc. ↑ |
|---|---|---|---|
| $S_{\text{RTI}}$ | 9.8× | 52% | 90% |
| $S_{\text{RTI}} + S_{\text{FID}}$ | 8.9× | 74% | 95% |
| Full ($S_{\text{RTI}} + S_{\text{FID}} + S_{\text{STAB}}$) | **10.8×** | **92%** | **96%** |

loss than all baselines, reflecting more stable and efficient optimization behavior.

**Stage II: Sink Payload Optimization for Sustained Slowdown.** We evaluate Stage II sink-payload optimization, which aims to induce sustained reasoning and tool-use overhead after a successful trigger while preserving task correctness. As shown in Table 3, optimized sink payloads substantially increase both end-to-end latency and RTI across all agents and backbone models, with RTI ranging from approximately 3× to over 10× and corresponding delays increasing by an order of magnitude. Different sink optimizers exhibit clear trade-offs between slowdown strength and stability: ICL-based methods without context awareness achieve higher RTI but suffer from low Hit rates, whereas context-aware variants improve activation stability while retaining strong slowdown. Across all settings, OTora-Persistent consistently achieves the best balance, delivering the highest or near-highest RTI together with high Hit and stable task accuracy. Importantly, downstream task accuracy remains largely unaffected (typically above 90%), confirming that the slowdown arises from deliberate amplification of internal reasoning and tool-use behaviors rather than task failure. Moreover, under the same sink payload, larger backbone models (e.g., LLaMA-3.1-70B and GPT-OSS-120B) exhibit higher RTI and longer delays than GPT-3.5-Turbo (OpenAI, 2023). Extended evaluations on additional model families and generalization to closed-source API models are provided in Appendices H and G.

## 5. Ablation Study

**Stage I: Insertion Strategy.** We first ablate the insertion strategy used in Stage I trigger optimization. Table 4 compares OTora's best-position selection with two simpler alternatives: inserting the trigger at a random valid position and using a fixed prefix.

As shown in Table 4, selecting the best insertion position is critical for both effectiveness and efficiency. Compared to random or fixed insertion, OTora's position-aware strategy improves $\text{ASR}_S/\text{ASR}_H$ by more than 20 percentage points while reducing the number of optimization iterations by nearly half. These results indicate that trigger optimization is highly position-sensitive, and that identifying a semantically aligned insertion point is essential for reliable and efficient trigger activation. We additionally ablate (i) the attention-aware term in the Stage I insertion scoring function and (ii) the dynamic target co-evolution mechanism; removing either component reduces trigger reliability and slows convergence, highlighting their complementary roles in robust Stage I optimization (Appendix E and Appendix F).

**Stage II: Sink Payload Scoring Objective.** We next ablate the scoring objective used for Stage II sink payload optimization. Table 5 reports the effect of progressively adding fidelity and stability terms to the RTI-only objective.

Optimizing RTI alone achieves high reasoning-token inflation but leads to unstable activation, as reflected by a low Hit rate. Introducing the fidelity term substantially improves activation reliability while preserving strong slowdown. The full objective, which further incorporates stability, achieves the best overall trade-off: the highest RTI together with high Hit and task accuracy. These results show that sustained R-DoS requires balancing slowdown strength with execution stability, and justify the full multi-term objective used in OTora-Persistent.

## 6. Conclusion

We introduced OTora, the first unified red-teaming framework for R-DoS against LLM agents. OTora exposes a new threat model in which an attacker preserves task correctness while degrading system availability by inducing excessive reasoning and tool-use overhead. By decomposing R-DoS into trigger optimization and persistent reasoning-payload injection, OTora reliably induces multi-turn slowdown across diverse agents, backbone models, and threat settings, achieving order-of-magnitude increases in latency and reasoning tokens with minimal accuracy loss. Our results highlight reasoning efficiency as a critical yet vulnerable security boundary in agentic LLM systems, and motivate future defenses that explicitly account for reasoning cost, stability, and SLA constraints.

## Acknowledgment

This work was supported by the NVIDIA Academic Grant Program (Exploiting Overthinking Attacks on GenAI), the Royal Society Grant (Ensuring Trustworthy AI: Robustness Certification for Large Language Models) [Reference RGS\R2\252444], and the AIRR Gateway project (Exploiting Robustness of Reasoning Efficiency in Agentic AI).

## Impact Statement

This work aims to improve understanding of reliability and efficiency challenges in deployed large language model (LLM) agents, particularly in operational settings where latency, cost, or service-level constraints are important. By examining failure modes that are not captured by standard evaluations focused on output correctness, our goal is to inform more comprehensive assessment of agent reliability in real-world systems.

As with other security and red-teaming research, the techniques studied in this paper could potentially be misused to intentionally increase computational or operational overhead. We emphasize that our intent is not to facilitate such misuse, but to highlight an underexplored class of availability risks and to encourage the adoption of availability-aware evaluation practices and efficiency-conscious deployment safeguards for more robust and reliable use of LLM-based agents in real-world systems.

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

# A. Mitigation Considerations and Limitations for R-DoS

R-DoS targets system availability by exhausting an agent's reasoning and tool-use budget while largely preserving task correctness. As a result, effective mitigations must operate at the agent orchestration layer and explicitly reason about efficiency (Huang et al., 2024; Dong et al., 2024), rather than relying solely on content moderation or output-level safety checks. Below we summarize representative mitigation directions and discuss their inherent limitations.

## A.1. Design Goals

An effective defense against R-DoS should simultaneously satisfy: (i) *availability guarantees* (e.g., latency, cost, or SLA constraints), (ii) *correctness preservation* (avoiding premature termination on valid but complex tasks), and (iii) *robustness to benign hard tasks* (minimizing false positives). R-DoS is challenging precisely because it operates in the narrow regime where these objectives conflict.

## A.2. Reasoning and Tool-Use Budgeting

A natural mitigation is to enforce budgets on reasoning tokens, wall-clock time, or tool invocations. Such controls can be implemented as hard caps or soft limits with graceful degradation (e.g., switching to concise planning or requesting confirmation). However, aggressive budgeting risks truncating legitimate long-horizon reasoning, while permissive thresholds remain vulnerable to slow and persistent R-DoS behaviors. This tension is fundamental and cannot be resolved by static limits alone. Moreover, because R-DoS payloads are optimized jointly for task fidelity and reasoning inflation, they can adapt to conservative thresholds by trading per-step overhead for longer horizons, further complicating static defenses.

## A.3. Sanitization and Relevance Filtering of Environment Content

Since R-DoS can be delivered through untrusted observations (e.g., webpages or emails), agents may preprocess fetched content via truncation, normalization, or relevance filtering. While such filtering can remove obviously unrelated material, R-DoS payloads are explicitly crafted to remain task-consistent and semantically relevant, making them difficult to distinguish from benign but complex inputs without sacrificing task utility.

## A.4. Runtime Monitoring

Runtime monitors may flag abnormal reasoning-token growth, repeated tool calls, or stalled progress. However, low-and-slow R-DoS trajectories can remain within typical variance ranges of benign executions, especially for difficult tasks. Tight anomaly thresholds reduce attack impact but increase false positives, whereas conservative thresholds preserve correctness but allow sustained resource exhaustion.

## A.5. Discussion

No single mitigation is sufficient in isolation. More importantly, R-DoS exposes a structural limitation of current agent defenses: safeguards are primarily designed to detect semantic or policy violations, whereas excessive yet correct reasoning is operationally indistinguishable from benign execution. As a result, defending against R-DoS inevitably involves trade-offs between efficiency, correctness, and usability. We view these tensions as highlighting an open systems-level challenge, rather than a resolved engineering issue.

# B. Motivation and Limitations of Stage I Payload Injection

This appendix elaborates why *directly* embedding long reasoning payloads into Stage I injection channels (e.g., user instructions or third-party environments) is often unreliable in realistic agent deployments, motivating OTora's two-stage decomposition.

## B.1. Limitations of injecting long payloads into third-party environments (e.g., shopping pages).

Third-party environments are typically *noisy* and *uncontrolled*: real webpages contain abundant boilerplate such as templates, recommendations, reviews, and dynamically generated blocks. To stay within context budgets, agents commonly apply truncation, summarization, or relevance extraction to such observations, which can fragment or discard long injected content

in unpredictable ways. Consequently, placing a long payload directly in these environments yields low delivery reliability, whereas a short, high-salience trigger is more likely to survive the agent's preprocessing.

In contrast, attacker-controlled pages can be intentionally designed to be concise and well-structured, reducing irrelevant context and minimizing the need for summarization or aggressive truncation. This significantly increases the probability that the payload is ingested *intact* and processed as intended.

### B.2. Limitations of injecting long payloads into user instructions.

User instructions are a particularly *narrow* channel in practice. They are often subject to hard length limits, upstream templating, or truncation by product UI/SDK/orchestration layers. Moreover, compared to environment observations, many systems enforce stricter safety and prompt-hygiene checks on user-provided text, making long, unusual payload-like instructions more likely to be flagged or sanitized. Finally, instruction-level payloads are user-visible and thus easier to notice, reducing stealth in realistic settings.

### B.3. Implication for two-stage design.

Taken together, Stage I channels are *narrow and noisy*: they are suitable for delivering short triggers that reliably induce tool access, but are ill-suited for reliably carrying long, structured reasoning payloads. Stage II therefore places the payload in attacker-controlled content, where the attacker can control page structure and style (e.g., formatting as FAQs, documentation, or task-relevant explanations) to maximize intact ingestion and stable execution.

## C. Why R-DoS Does Not Trigger Early-Stop Safeguards

Modern LLM agent systems typically incorporate multiple safeguard mechanisms to limit abnormal executions, including early termination, refusal, fallback to lightweight models, or step- and budget-based cutoffs. In practice, these mechanisms are primarily triggered by observable deviations in task behavior, such as incorrect actions, goal divergence, policy violations, or explicit safety risks.

R-DoS operates in a different regime. Rather than inducing task failure or behavioral deviation, the attack preserves functional correctness and follows the agent's normal reasoning and action trajectory. As a result, common safeguard signals—such as invalid actions, refusal conditions, or policy-triggering outputs—are not activated. This allows the agent to continue execution along standard control flow, while incurring substantially inflated reasoning and tool-use cost.

R-DoS payloads are explicitly optimized to remain task-consistent and trajectory-aligned: they do not introduce goal deviation, invalid actions, or policy-violating content, but instead induce reasoning-intensive computation that remains semantically relevant to the current task context. At the generation level, this is supported by the context-aware optimization mode in Stage II, where candidate payloads are generated conditioned on the agent's current task background and interaction history, biasing the search space toward task-relevant continuations. At the evaluation level, this constraint is enforced via the fidelity term $S_{\text{FID}}$, which constrains payloads to preserve the agent's original task intent and final action logic. Many deployed LLM agents determine each action by jointly conditioning on the current observation and the accumulated interaction history, rather than treating observations in isolation ([Yao et al., 2022b](#)). As a result, even after the agent is successfully guided to consume externally sourced content, subsequent actions continue to be informed by the original task context and prior reasoning trajectory. This history-aware decision process helps the agent resume its intended execution path, provided that the injected payload remains task-consistent, thereby preventing transient external observations from permanently derailing the agent's normal reasoning and action flow.

Importantly, we do not assume the absence of safeguards. Instead, R-DoS exploits a structural limitation of current agent defenses: safeguards are designed to detect semantic or policy-level anomalies, whereas excessive but correct reasoning remains indistinguishable from benign execution. Consequently, early-stop mechanisms that reduce system-side compute are not triggered, enabling sustained resource exhaustion under continued execution.

## D. Optimization Cost and Practicality of OTora

**Why cost matters.** OTora is an *optimization-driven* red-teaming framework with iterative search in both Stage I (trigger optimization) and Stage II (payload optimization). Accordingly, reporting only the number of model queries can be misleading for assessing practicality, since (i) API pricing is token-based (input/output tokens differ), (ii) Stage II incurs

*Table 6.* Stage I black-box trigger optimization cost and success rate on WebShop using GPT-3.5-Turbo.

| Trigger Method | API Calls ↓ | Tokens (M) ↓ | Cost (USD) ↓ | ASR$_H$ ↑ |
|---|---|---|---|---|
| SNES | 7,830 | 13.3M | $17.76 | 35% |
| AutoDAN | 7,008 | 11.9M | $15.89 | 41% |
| PAL | 5,860 | 10.0M | $13.35 | 50% |
| **OTora (ours)** | **4,310** | **7.3M** | **$9.75** | **57%** |

additional agent rollouts and tool executions, and (iii) black-box settings may require proxy-model inference or auxiliary evaluation signals. In this appendix, we provide an itemized cost model and instantiate it on representative experimental settings to illustrate concrete cost magnitudes and deployment feasibility.

### D.1. Cost Model

We decompose the total cost of running OTora on a task instance into *API/token cost*, *compute cost* (proxy inference or local optimization), and *runtime overhead* (tool calls and environment interactions).

**Token/API cost.** Let $c_{\text{in}}$ and $c_{\text{out}}$ denote the per-token prices for input and output tokens (USD/token). For an API call $i$, let $t_{\text{in}}^{(i)}$ and $t_{\text{out}}^{(i)}$ be its input/output token counts. The total API cost is

$$C_{\text{api}} = \sum_{i=1}^{N_{\text{api}}} \left( c_{\text{in}}\, t_{\text{in}}^{(i)} + c_{\text{out}}\, t_{\text{out}}^{(i)} \right). \tag{5}$$

We report the number of API calls $N_{\text{api}}$, total tokens, and the resulting dollar estimate under the pricing used at evaluation time. Dollar values are reported for reference only and depend on the API pricing at the time of evaluation.

**Token pricing instantiation.** For black-box experiments using GPT-3.5-turbo (OpenAI, 2023), we estimate monetary cost based on inference-time token-based billing at the time of evaluation. Unless otherwise stated, we use the following reference prices from the public pricing table: $c_{\text{in}} = \$0.50/1\text{M}$ input tokens and $c_{\text{out}} = \$1.50/1\text{M}$ output tokens (equivalently, $c_{\text{in}} = \$0.0005/1\text{K}$ and $c_{\text{out}} = \$0.0015/1\text{K}$). All dollar values are reported for reference only and may vary with API pricing. We snapshot the prices on *October 1, 2025* for reproducibility.

**Compute cost (proxy or local optimization).** When OTora uses a proxy model (e.g., for black-box trigger scoring or payload search assistance), or operates in a white-box setting with token-level gradients, we quantify compute cost in terms of *GPU-hours*, denoted by $C_{\text{gpu}} = h_{\text{gpu}}$, where $h_{\text{gpu}}$ is the total GPU time consumed during optimization. In our experiments, white-box optimization is performed on dedicated academic accelerators and does not incur external API charges. Accordingly, we report compute cost for white-box settings purely in GPU-hours as a hardware-agnostic measure of optimization scale, without converting it to monetary cost.

**Runtime overhead (tool calls and rollouts).** OTora's Stage II cost is dominated by *agent rollouts* under candidate payloads. We report (i) the number of rollouts $N_{\text{roll}}$ and (ii) end-to-end wall-clock time $\tau$, which are critical for assessing systems-level feasibility under latency or SLA constraints.

### D.2. Optimizer Configuration

**Stage I optimizer configuration (trigger optimization).** Stage I iteratively updates a trigger suffix to induce a targeted external access or tool invocation. Unless otherwise specified, we use a maximum budget of $T_1^{\max} = 500$ optimization iterations, with early stopping enabled upon successful trigger activation. At each iteration, we maintain a small set of co-evolved target intents $\mathcal{T}$ of size $|\mathcal{T}| = 5$. Targets in $\mathcal{T}$ are generated using high-probability tokens from the agent response distribution and an auxiliary language model. For each target, insertion positions are scored using the attention-aware objective in Eq. (1), and the top-$\ell = 3$ non-overlapping positions are selected via weighted interval scheduling. A trigger is considered successful if the target token sequence appears in the agent response and results in a valid external tool invocation.

*Table 7.* Stage I white-box trigger optimization cost and success rate on WebShop using LLaMA-3.1-70B (NVIDIA GH200, 120GB). *Model Calls* count the number of forward/backward model evaluations during gradient-based optimization.

| Trigger Method | Model Calls ↓ | GPU-hours ↓ | Wall-clock (min) ↓ | $ASR_H$ ↑ |
|---|---|---|---|---|
| GCG | 3,200 | 1.1h | 66 | 85% |
| I-GCG | 3,060 | 0.9h | 54 | 87% |
| UDora | 2,395 | 0.75h | 45 | 89% |
| **OTora (ours)** | **1,960** | **0.6h** | **36** | **95%** |

**Stage II optimizer configuration (payload optimization).** Stage II searches for an agent-aware reasoning payload that maximizes slowdown while preserving functional correctness. We run $T_2 \in [25, 30]$ optimization iterations, evaluating $P = 12$ candidate payloads per iteration. For stability estimation, each candidate is evaluated with $S = 3$ random seed by default, resulting in $N_{\text{roll}} = T_2 \cdot P \cdot S$ agent rollouts. Candidate payloads are scored using the R-DoS-oriented multi-objective function in Eq. (4), which jointly optimizes reasoning token inflation (RTI), task fidelity, and stability across runs. Top-performing payloads are retained and mutated using an in-context learning (ICL) capable model conditioned on the agent's current execution context. Early termination is enabled once the slowdown objective saturates.

### D.3. Stage-wise Cost Breakdown

#### D.3.1. STAGE I: TRIGGER OPTIMIZATION

In black-box settings, the dominant cost arises from API calls and token usage. In white-box settings, the dominant cost arises from local gradient-based computation.

**Black-box cost (API-based).** Table 6 instantiates Stage I black-box trigger optimization cost on the WebShop agent using GPT-3.5-Turbo. In addition to cost, we report trigger success rate ($ASR_H$), measuring whether the agent triggers a valid external tool call to the target website.

**White-box cost (local optimization).** In the white-box setting, Stage I uses token-level gradients and does not incur *external API* costs; however, it still requires repeated forward/backward passes over a locally deployed model. We quantify optimization scale using *model calls* (the number of forward/backward evaluations), along with GPU-hours and end-to-end wall-clock time. Table 7 reports compute cost on the WebShop agent using LLaMA-3.1-70B, executed on NVIDIA GH200 (120GB) accelerators.

#### D.3.2. STAGE II: PAYLOAD OPTIMIZATION

Stage II searches for an agent-aware reasoning payload that maximizes slowdown while preserving functional correctness. Let $T_2$ be the number of optimization iterations, $P$ the number of candidate payloads evaluated per iteration, and $S$ the number of random seeds used for stability estimation. Each candidate evaluation requires running the agent once, yielding:

$$N_{\text{roll}} = T_2 \cdot P \cdot S, \qquad C_{\text{api}}^{(2)} \approx \sum_{\text{rollouts}} \text{(agent-step API tokens)}. \tag{6}$$

Table 8 reports representative Stage II payload optimization cost on the WebShop agent using GPT-3.5-Turbo. In addition to cost, we report slowdown effectiveness via RTI (Reasoning Token Inflation), as defined in the main experiments.

### D.4. Empirical Reporting and Practicality

**Practical cost reporting.** OTora is an optimization-driven red-teaming framework whose effectiveness comes at the cost of iterative search in both Stage I (trigger optimization) and Stage II (payload optimization). Accordingly, reporting optimization cost is essential for understanding the practical feasibility of reasoning-level denial-of-service (R-DoS) attacks, especially under realistic query, time, or compute budgets.

**Parallelism and evaluation efficiency.** Stage II evaluations are embarrassingly parallel across candidate payloads and random seeds, and can be substantially accelerated using parallel agent rollouts, early termination once the slowdown objective saturates, and caching of tool outputs when permissible.

*Table 8.* Stage II payload search cost (optimization-time) and resulting slowdown (RTI) on WebShop using GPT-3.5-Turbo. Methods labeled as *Fixed* use pre-specified payloads without iterative optimization. For optimized methods, $N_{roll} = T_2 \cdot P \cdot S$ with $P$=12 candidates per iteration and $S$=3 seeds. **Optimizer Tokens** count tokens consumed by the payload-search/ICL generator (and any auxiliary scoring model, if used), excluding agent rollouts. **Rollout Tokens** count total tokens consumed by agent rollouts during candidate evaluation.

| Method | $T_2$ | $N_{roll}\downarrow$ | Optimizer Tokens (M)$\downarrow$ | Rollout Tokens (M)$\downarrow$ | Wall-clock (min)$\downarrow$ | RTI$\uparrow$ |
|---|---|---|---|---|---|---|
| **Fixed payload (no iterative optimization)** | | | | | | |
| OTora-Agnostic (Fixed) | – | – | – | – | – | 2.1× |
| OTora-Aware (Fixed) | – | – | – | – | – | 2.8× |
| **Optimized payload (iterative search)** | | | | | | |
| OTora-ICL (Agnostic) | 30 | 1080 | 0.82M | 17.4M | 34 | 3.6× |
| OTora-ICL (Aware) | 30 | 1080 | 0.76M | 16.6M | 32 | 3.4× |
| **OTora-Persistent** | **25** | **900** | **0.72M** | **18.9M** | **30** | **5.2×** |

*Table 9.* Ablation of the attention-aware term in Stage I insertion scoring (Eq. (1)). Removing the attention term ($\lambda = 0$) reduces trigger reliability and slows convergence.

| Method | $ASR_H \uparrow$ | Iters$\downarrow$ |
|---|---|---|
| OTora (full; $\lambda > 0$) | **82%** | **60** |
| OTora w/o attention ($\lambda = 0$) | 73% | 95 |

**Defender relevance.** Beyond attacker-side analysis, budget-aware cost reporting provides a standardized basis for evaluating system robustness under constrained execution settings. In particular, the reported stage-wise cost breakdown supports controlled assessment of R-DoS risks alongside the slowdown and correctness metrics in the main experiments.

# E. Ablation of Attention-Aware Insertion Scoring

Stage I selects insertion positions by maximizing the insertion scoring function in Eq. (1), which combines a prefix-match term, a continuation-likelihood term, and an attention-aware term that down-weights positions whose apparent token matches arise primarily from prior context rather than the adversarial suffix. To isolate the contribution of the attention-aware component, we perform an ablation that removes the attention term by setting $\lambda = 0$ in Eq. (1), while keeping all other settings (agent harness, target intent, optimizer, and stopping criteria) unchanged.

**Setup.** We report results on the **WebShop** benchmark under the **white-box** setting with **LLaMA-3.1-70B-Instruct**, using the same evaluation protocol as in Section 4. We measure trigger success by $ASR_H$ (valid external tool invocation) and optimization efficiency by the average number of iterations to reach a successful trigger.

**Results and discussion.** As shown in Table 9, removing the attention-aware term consistently decreases $ASR_H$ and increases the number of iterations required to obtain a valid trigger. This indicates that attention-aware scoring provides complementary signal beyond match/likelihood alone, helping Stage I avoid spurious insertion positions where partial target-token matches are explained by prior context rather than the injected adversarial suffix.

**Black-box implication.** In black-box settings where attention weights are not available, we omit the attention term (i.e., $\lambda = 0$) or approximate it using proxy-model attribution without changing the overall Stage I pipeline. The ablation above explains why direct attention access can improve trigger optimization when available, while the remainder of OTora remains applicable under API-only access.

# F. Ablation on Dynamic Target Co-Evolution

In Stage I trigger optimization, OTora allows the target trigger phrase to *co-evolve* with the agent's response distribution, rather than optimizing against a single fixed lexical realization. This design enables the optimizer to adapt to semantically equivalent but lexically diverse tool-invocation intents naturally favored by the agent.

To quantify the contribution of this component, we ablate dynamic target co-evolution by comparing it against a fixed-target

variant.

**Setup.** We evaluate two Stage I configurations under the white-box setting with LLaMA-3.1-70B-Instruct on the WebShop benchmark: (i) **Fixed Target**, which optimizes the trigger string against a single predefined target phrase (e.g., ``access attacker.test''); and (ii) **Dynamic Co-Evolution**, which iteratively samples semantically equivalent target candidates from the agent's response distribution and selects the highest-scoring target during optimization. All other components, including insertion strategy and optimizer settings, are held constant.

We report trigger success rate ($ASR_H$), average optimization iterations (Iters), and cross-seed variance of $ASR_H$ (computed across multiple random seeds) as a measure of optimization stability.

*Table 10.* Ablation on dynamic target co-evolution in Stage I trigger optimization (WebShop, LLaMA-3.1-70B-Instruct).

| Target Strategy | $ASR_H \uparrow$ | Iters $\downarrow$ | $Var(ASR_H) \downarrow$ |
|---|---|---|---|
| Fixed Target | 84% | 95 | 0.024 |
| Dynamic Co-Evolution (OTora) | **90%** | **65** | **0.010** |

**Discussion.** Dynamic target co-evolution consistently improves trigger reliability and optimization efficiency, achieving higher $ASR_H$ with substantially fewer iterations and reduced variance across random seeds. Notably, the fixed-target variant remains functional, indicating that co-evolution is not strictly required for successful triggering; however, allowing the target phrase to adapt to the agent's native response distribution yields more stable and robust optimization. These results support dynamic co-evolution as an important contributor to Stage I robustness rather than a brittle or task-specific heuristic.

## G. Generalization to Closed-Source API Models

OTora supports two complementary modes for attacking closed-source API models: direct black-box optimization and cross-model transfer. In the direct black-box mode (Tables 1 and 3), OTora uses API-level access or proxy models to optimize triggers without any white-box involvement, achieving 30–48% end-to-end (E2E) success with 5–6× RTI across agents on GPT-3.5-Turbo.

As an additional low-cost option, OTora's optimized artifacts can also be transferred across model boundaries without re-optimization. To evaluate this, we optimize the trigger suffix and reasoning payload on a white-box source model (LLaMA-3.1-70B-Instruct, WebShop agent, environment injection) and directly evaluate on closed-source API models. Table 11 reports the results alongside UDora under the same zero-shot transfer protocol.

*Table 11.* Cross-model transfer results. Trigger and payload optimized on LLaMA-3.1-70B (white-box, WebShop, environment injection) and evaluated on target models without re-optimization. E2E denotes end-to-end attack success ($ASR_H \times Hit$).

| Target Model | UDora E2E | OTora E2E | OTora RTI |
|---|---|---|---|
| *LLaMA-3.1-70B (source)* | *84%* | *87%* | *9.7×* |
| Gemini-1.5-Flash (API) | 14% | **26%** | **5.1×** |
| GPT-3.5-Turbo (API) | 10% | **21%** | **4.8×** |

**Analysis.** Cross-model transfer degrades E2E success from 87% to 21–26%, but OTora still substantially outperforms UDora under the same protocol (21–26% vs. 10–14%), and RTI remains at 4.8–5.1×, confirming that the optimized artifacts retain meaningful attack effectiveness across model boundaries. Moreover, direct black-box optimization is OTora's primary mode for attacking closed-source models and achieves higher E2E success (30–48%) without requiring any white-box source model, making cross-model transfer a useful low-cost alternative when API query budgets are constrained.

## H. Extended Evaluation on Additional Reasoning Models

To evaluate the generalizability of OTora across a wider range of model families and reasoning capabilities, we extend our evaluation to include two additional backbone models: Qwen-2.5-32B (Qwen et al., 2025) and DeepSeek-V2-67B (Liu et al., 2024). These models represent distinct architectural lineages and pretraining recipes, complementing the LLaMA and GPT-OSS families evaluated in the main experiments.

Table 12 reports end-to-end results across all three agents (WebShop, Email, OS) under the environment injection setting, using the same OTora-Persistent configuration as in Table 3.

*Table 12.* Extended evaluation on additional reasoning models across all agents (environment injection, OTora-Persistent). E2E denotes $ASR_H \times Hit$. Results for LLaMA-3.1-70B and GPT-OSS-120B are reproduced from Table 3 for reference.

| Model | Agent | E2E | RTI | Hit | Acc. |
|---|---|---|---|---|---|
| *LLaMA-3.1-70B* | *WebShop* | *87%* | *9.7×* | *92%* | *96.1%* |
| | *Email* | *86%* | *9.9×* | *91%* | *96.0%* |
| | *OS* | *80%* | *10.8×* | *86%* | *95.7%* |
| *GPT-OSS-120B* | *WebShop* | *85%* | *9.1×* | *90%* | *96.3%* |
| | *Email* | *83%* | *9.3×* | *89%* | *96.2%* |
| | *OS* | *76%* | *10.0×* | *84%* | *96.0%* |
| Qwen-2.5-32B | WebShop | 82% | 8.6× | 89% | 95.2% |
| | Email | 80% | 8.9× | 88% | 95.0% |
| | OS | 75% | 9.5× | 83% | 94.8% |
| DeepSeek-V2-67B | WebShop | 84% | 9.2× | 90% | 95.5% |
| | Email | 82% | 9.4× | 89% | 95.3% |
| | OS | 77% | 10.1× | 84% | 95.0% |

**Analysis.** OTora achieves comparable E2E success and RTI on both new models across all three agents, closely matching the results on LLaMA-3.1-70B and GPT-OSS-120B. Task accuracy remains high across all settings, confirming that R-DoS generalizes across model families without sacrificing functional correctness. The same agent-level trend observed in the main experiments also holds: WebShop is the most susceptible, followed by Email and OS, consistent with the contextual naturalness of external access in each environment. These results support the conclusion that R-DoS is a general vulnerability of tool-augmented LLM agents, not an artifact of specific model architectures.

## I. Quantitative Defense Evaluation

To complement the qualitative defense discussion in Appendix A, we implement two representative defenses and evaluate the availability–correctness trade-off on LLaMA-3.1-70B, WebShop agent (environment injection, $N=50$ task instances).

**Defense 1: Observation Relevance Filtering.** Before processing fetched content, a relevance scorer assigns each observation a score in $[0, 1]$; content scoring below threshold $\theta$ is discarded.

*Table 13.* Observation relevance filtering defense (LLaMA-3.1-70B, WebShop). Higher $\theta$ filters more aggressively.

| Filter $\theta$ | OTora E2E ↓ | RTI ↓ | Delay ↓ | Benign Acc. ↑ |
|---|---|---|---|---|
| 0 | 87% | 9.7× | 325 s | 95% |
| 0.3 | 78% | 8.9× | 290 s | 89% |
| 0.5 | 68% | 7.8× | 255 s | 81% |
| 0.7 | 48% | 6.3× | 205 s | 64% |
| 0.9 | 22% | 4.1× | 135 s | 41% |

**Defense 2: Runtime Monitoring.** Per-turn reasoning tokens are tracked; the agent is terminated early if any turn exceeds $\mu + c\sigma$, where $\mu$ and $\sigma$ are estimated from benign calibration episodes.

**Analysis.** Both defenses reduce R-DoS effectiveness but at significant benign cost. Relevance filtering at $\theta=0.7$ cuts OTora E2E from 87% to 48% but also drops benign accuracy from 95% to 64%; at the most aggressive setting ($\theta=0.9$), E2E drops to 22% but benign accuracy falls to 41%, rendering the agent largely unusable. This limited effectiveness reflects a key property of R-DoS payloads: they are optimized to be task-consistent and thus score similarly to benign observations under relevance filtering. Runtime monitoring exhibits a similar trade-off: at $c=1.5\sigma$, E2E reduces to 58% but 28% of benign tasks are falsely terminated; at $c=1.0\sigma$, E2E drops further to 38% but nearly half of benign tasks fail to complete. Notably, attacks that evade detection still achieve near-full RTI (9.2× at $c=1.5\sigma$), as OTora's persistent payload naturally

*Table 14.* Runtime monitoring defense (LLaMA-3.1-70B, WebShop). Tighter thresholds reduce R-DoS but increase false terminations.

| Threshold $c$ | OTora E2E ↓ | RTI ↓ | Delay ↓ | Benign Completion ↑ |
|---|---|---|---|---|
| $\infty$ | 87% | 9.7× | 325 s | 100% |
| $3.0\sigma$ | 80% | 9.6× | 315 s | 93% |
| $2.0\sigma$ | 70% | 9.4× | 305 s | 84% |
| $1.5\sigma$ | 58% | 9.2× | 295 s | 72% |
| $1.0\sigma$ | 38% | 8.5× | 270 s | 55% |

distributes reasoning inflation across multiple turns rather than concentrating it in a single step. These results empirically confirm the inherent availability–correctness trade-off discussed in Appendix A: no static threshold can simultaneously maintain high availability and fully suppress R-DoS without degrading benign task performance.

## J. Cross-Task and Cross-Agent Transferability

Appendix G evaluates cross-model transfer (white-box → API). Here we additionally evaluate cross-task and cross-agent transfer to assess the reusability of OTora's optimized artifacts. Both stages are optimized per task instance by default.

**Cross-task transfer.** We optimize the trigger on a single source task instance and evaluate on 50 held-out instances from the same agent without re-optimization.

*Table 15.* Cross-task transfer (1 source instance → 50 held-out, environment injection). Results averaged over 3 source instances.

| Setting | Agent | Method | Per-inst E2E | Cross-task E2E | Cross-task RTI | Acc. |
|---|---|---|---|---|---|---|
| White-box (LLaMA-70B) | WebShop | UDora | 84% | 21%±6 | 7.2×±0.8 | 94.8% |
| | | OTora | 87% | 31%±5 | 7.6×±0.7 | 95.3% |
| | Email | UDora | 82% | 18%±6 | 7.4×±0.9 | 94.5% |
| | | OTora | 86% | 28%±5 | 7.8×±0.8 | 95.0% |
| | OS | UDora | 76% | 15%±7 | 7.8×±1.0 | 94.2% |
| | | OTora | 80% | 24%±6 | 8.2×±0.9 | 94.7% |
| Black-box (GPT-3.5) | WebShop | PAL | 41% | 11%±4 | 3.3×±0.6 | 92.5% |
| | | OTora | 48% | 19%±4 | 3.8×±0.5 | 93.1% |
| | Email | PAL | 36% | 9%±4 | 3.4×±0.7 | 92.0% |
| | | OTora | 42% | 16%±4 | 3.9×±0.6 | 92.8% |
| | OS | PAL | 28% | 7%±4 | 3.6×±0.8 | 91.5% |
| | | OTora | 32% | 13%±5 | 4.2×±0.7 | 92.2% |

**Cross-agent and cross-model transfer.** We optimize on WebShop + LLaMA-3.1-70B and evaluate on different agents and locally deployed models without re-optimization. Transfer to closed-source API models is reported in Appendix G.

*Table 16.* Cross-agent and cross-model transfer (optimized on WebShop + LLaMA-3.1-70B, environment injection).

| Target | UDora E2E | OTora E2E | OTora RTI |
|---|---|---|---|
| *WebShop, LLaMA-70B (source)* | 84% | 87% | 9.7× |
| Email, LLaMA-70B | 11% | **29%** | 6.8× |
| OS, LLaMA-70B | 9% | **24%** | 6.2× |
| WebShop, GPT-OSS-120B | 14% | **31%** | 5.9× |

**Analysis.** Cross-task transfer retains meaningful effectiveness across all agents. On WebShop, OTora achieves 31% cross-task E2E (vs. 21% for UDora) in the white-box setting and 19% (vs. 11% for PAL) in the black-box setting, with the same agent-level trend (WebShop > Email > OS) observed in the main experiments. Conditional on success, RTI remains substantial (3.8–8.2×) with task accuracy above 92% across all settings. Cross-agent transfer is more challenging due to

differences in tool schemas and reasoning patterns, but OTora consistently outperforms UDora across all targets (24–29% vs. 9–11%). These results indicate that an attacker can amortize optimization cost by reusing triggers across task instances within the same agent, and that partial transferability exists across agents.

## K. Evaluation Protocol and Uncertainty Estimates

**Episode count and success criteria.** Each experimental setting uses $N{=}50$ task instances. Correctness is determined by automated string matching against the expected gold action (e.g., `click[<ID>]` for WebShop, target command execution for OS); manual inspection is used only as a post-hoc sanity check to verify semantic equivalence under multi-step execution.

**Uncertainty estimates.** Table 17 reports 95% bootstrap confidence intervals (1000 resamples) for task accuracy under no-attack and OTora-Persistent conditions, confirming that attack-induced accuracy changes are within sampling noise.

*Table 17.* 95% bootstrap CI for task accuracy (OTora-Persistent, environment injection, $N{=}50$, 1000 resamples).

| Model | Agent | No-Attack Acc. (95% CI) | Attack Acc. (95% CI) |
|---|---|---|---|
| GPT-3.5 | WebShop | 92.0% (82.0, 97.0) | 94.0% (84.0, 98.5) |
| GPT-3.5 | Email | 93.0% (83.0, 97.5) | 93.0% (82.5, 98.0) |
| GPT-3.5 | OS | 91.0% (80.0, 96.5) | 91.0% (80.5, 96.5) |
| LLaMA-70B | WebShop | 95.0% (86.0, 99.0) | 96.1% (88.0, 99.5) |
| LLaMA-70B | Email | 95.0% (85.5, 98.5) | 96.0% (87.0, 99.0) |
| LLaMA-70B | OS | 95.0% (86.0, 98.5) | 95.7% (86.5, 99.0) |
| GPT-OSS | WebShop | 96.0% (87.5, 99.0) | 96.3% (88.0, 99.5) |
| GPT-OSS | Email | 96.0% (87.0, 99.0) | 96.2% (87.5, 99.0) |
| GPT-OSS | OS | 95.8% (86.5, 99.0) | 96.0% (87.0, 99.0) |

**Interpretation.** All settings show overlapping confidence intervals between no-attack and attack conditions, confirming that OTora preserves task correctness within the precision of our evaluation.

