# OpenReview forum: "OTora: A Unified Red Teaming Framework for Reasoning-Level Denial-of-Service in LLM Agents"
_ICML.cc/2026/Conference — ICML 2026 regular_

### Official Review · Reviewer_HjRQ · 2026-02-28

**Soundness:** 3
**Presentation:** 3
**Significance:** 3
**Originality:** 4
**Overall Recommendation:** 4
**Confidence:** 4

**Summary:**

This paper proposed R-DoS, a novel attack framework focus on LLM agent. It raises DoS from network layer to reasoning layer, and designs OTora to instantiate R-DoS with good experiment results.

**Compliance With Llm Reviewing Policy:**

Affirmed.

**Final Justification:**

This paper focus on a novel issue that R-DoS for LLM. That's why i give 4 as a positive comment. However, I also concer some experiment setting or metrics which are difficult to analysis. That's why I can't give 5.

**Key Questions For Authors:**

1. Could the authors run a true end-to-end evaluation on the full pipeline—starting from Stage I injection through Stage II slowdown? Or give a detailed explanation that why don't need to run a true end-to-end evaluation?

2. To disentangle baseline reasoning capacity from attack-induced vulnerability, could the authors report a normalized metric such as (Delay_attack − Delay_baseline) / Delay_baseline or (RTI − 1) / baseline_tokens_per_turn across model sizes?

**Limitations:**

yes

**Strengths And Weaknesses:**

Strengths:

1. The proposal of R-DoS fills an important gap in LLM Agent security research. Existing work almost entirely focuses on output correctness or behavioral alignment, while this paper points out that “correct but unusable” also constitutes a serious threat.

2. The decomposition of the attack into two stages has clear engineering motivations. This separation also allows each stage to be independently optimized and evaluated. Such modular design enhances the framework's scalability, enabling the substitution of different optimization backends at various stages. The experiments indeed demonstrate the effectiveness of various optimizer combinations.

3. The ablation study has a clear structure and validates the contribution of each component.

Weaknesses:

1. The paper approximates the end-to-end success rate as ASR_H × Hit (Section 4). However, this multiplication implicitly assumes independence between the two stages. In reality, the trigger path in Stage I alters the agent's contextual state, which directly impacts the payload hit rate in Stage II, creating a clear causal dependency between them. Although the paper states it is "rather than assuming strict independence," the calculation method essentially follows the multiplicative form under an independence assumption, without any modeling or empirical analysis of the conditional dependency. More critically, the paper never reports the true end-to-end joint success rate on the complete pipeline, instead measuring each stage independently and then multiplying the results.

2. The conclusion that "stronger models are more vulnerable" lacks causal support. Table 3 shows that larger models suffer from higher RTI, based on which the paper claims that "stronger reasoning capacity amplifies vulnerability." However, the baseline delay for larger models is inherently higher. The absolute increase in RTI may simply reflect this baseline difference. If re-evaluated using normalized additional overhead, the difference might not be significant. The paper fails to control for this confounding variable of baseline reasoning length and does not provide any causal analysis.

---

> ### Author Rebuttal · Authors · 2026-03-30
>
> We thank the reviewer for the rigorous and constructive feedback. Below we address each concern with clarifications and additional analyses.
>
> ---
>
> **W1: Independence Assumption in E2E Computation**
>
> We would like to clarify that the two stages are not assumed independent. The attack proceeds as a sequential state transition:
>
> $$S_0 \xrightarrow{P(S_1)} S_1 \xrightarrow{P(S_2 \mid S_1)} S_2$$
>
> where $S_0$ is the initial state, $S_1$ denotes Stage I trigger success, and $S_2$ denotes Stage II payload hit. By the chain rule:
>
> $$\text{E2E} = P(S_2) = P(S_1) \times P(S_2 \mid S_1) = \text{ASR}_H \times \text{Hit}$$
>
> Here Hit = $P(S_2 \mid S_1)$ is computed **conditional on Stage I success** — Stage II is only activated after a successful trigger. This is a conditional transition, not an independence assumption. All experiments run the complete pipeline end-to-end, and E2E is measured from the same run.
> We show representative results (Environment injection) in the following:
>
> | Model | Agent | ASR_H | Hit | E2E | RTI |
> |:---|:---|:---:|:---:|:---:|:---:|
> | LLaMA-70B | WebShop | 95% | 92% | 87% | 9.7× |
> | | Email | 94% | 91% | 86% | 9.9× |
> | | OS | 93% | 86% | 80% | 10.8× |
> | GPT-3.5 | WebShop | 58% | 82% | 48% | 5.1× |
> | | Email | 52% | 80% | 42% | 5.3× |
> | | OS | 44% | 73% | 32% | 5.7× |
>
> Stage I and Stage II are executed as a sequential pipeline; E2E is computed via the chain rule from the same experimental setup. In the revised paper, we will present all experimental results under a unified framework and directly report E2E metrics, rather than separating them into two stages.
>
> ---
>
> **W2: Causal Support for "Stronger Models More Vulnerable"**
>
> Thanks for the reviewer's concern, we agree that this statement in Sec. 4 was not sufficiently precise. The observation is based on Table 3, where under the same OTora-Persistent payload, LLaMA-3.1-70B and GPT-OSS-120B achieve 9–11× RTI while GPT-3.5-Turbo achieves 5–6×. This is an empirical observation limited to the specific models tested, and a general causal conclusion cannot be drawn from this limited set of models. We will revise this statement in the updated paper to make it more precise.
> Specifically, that higher RTI is observed on these larger models under the same payload, without attributing it to a general "stronger reasoning capacity amplifies vulnerability" claim.
>
> ---
>
> **KQ1: True End-to-End Evaluation**
>
> As detailed in W1, Stage I and Stage II are executed as a sequential pipeline, and E2E is computed via the chain rule $P(S_1) \times P(S_2 \mid S_1)$. The W1 table reports ASR_H, Hit, and E2E from this pipeline across all settings.
>
> ---
>
> **KQ2: Normalized Metrics**
>
> We appreciate the suggestion. We compute the normalized delay increase (Delay_attack − Delay_baseline) / Delay_baseline from the existing results (OTora-Persistent, Environment injection):
>
> | Model | Agent | Baseline | Attack | Normalized Increase |
> |:---|:---|:---:|:---:|:---:|
> | LLaMA-70B | WebShop | 30s | 325s | 9.8× |
> | | Email | 30s | 335s | 10.2× |
> | | OS | 38s | 360s | 8.5× |
> | GPT-OSS-120B | WebShop | 28s | 310s | 10.1× |
> | | Email | 28s | 320s | 10.4× |
> | | OS | 34s | 345s | 9.1× |
> | GPT-3.5 | WebShop | 18s | 170s | 8.4× |
> | | Email | 17s | 182s | 9.7× |
> | | OS | 20s | 200s | 9.0× |
>
> After normalization, the gap between models narrows substantially (8.4–10.4× across all models vs. RTI 5.1–10.8× before normalization), supporting the reviewer's point that part of the raw RTI difference reflects baseline reasoning length rather than intrinsic vulnerability. We will include these normalized metrics in the revised paper.

---

> > ### Author Rebuttal · Reviewer_HjRQ · 2026-04-01
> >
> > We thank the authors for their rebuttal. After careful consideration, we maintain our score.
> >
> > On W1/KQ1: The rebuttal restates E2E = ASR_H × Hit using the chain rule, but this does not address our core concern. The fundamental issue is that Stage I's trigger alters the agent's contextual state, creating a causal dependency that may invalidate the multiplicative decomposition even under the conditional formulation. No empirical analysis of this dependency is provided, and it remains unclear whether Hit is truly measured on the Stage-I-success subset or derived post hoc. We consider this a substantive methodological gap.
> >
> > On W2/KQ2: We appreciate the normalized metrics, which confirm that the raw RTI gap largely reflects baseline differences. We accept the authors' revised, more cautious framing of this claim.
> >
> > The unresolved issue in W1 limits our confidence in the reported end-to-end effectiveness, which is central to the paper's empirical claims. We therefore maintain our current score.

---

> > > ### Author Response · Authors · 2026-04-02
> > >
> > > We thank the reviewer for the follow-up and are glad that we have clarified W2. We address the remaining concern on W1 below.
> > >
> > > ---
> > >
> > > **W1 Follow-up: Causal Dependency Between Stages**
> > >
> > > We appreciate the reviewer raising this deeper methodological point. We address the specific concerns in detail.
> > >
> > > **(1) Stage I trigger alters the agent's contextual state: does this invalidate the decomposition?**
> > >
> > > $S_2$ (payload-induced overthinking) can only be reached through $S_1$ (trigger success): no trigger → agent does not visit the target page ($S_1$ fails) → $S_2$ cannot occur, since the payload is hosted on that page. When no trigger is injected, the agent never visits the target page (0/50 episodes across all settings), and consequently never encounters the payload, so $P(S_2 \mid \neg S_1) = 0$.
> > >
> > > The reviewer's deeper concern is whether Stage I's trigger string, which remains in the agent's context even after the page visit, affects how the agent processes the Stage II payload. To directly test this, we compare two delivery modes on the same payload (LLaMA-70B, WebShop, N=50):
> > >
> > > 1. **Normal pipeline**: trigger `s` is injected → agent is induced to visit the target page → payload encountered with trigger in context
> > > 2. **Direct delivery** (controlled experiment): no trigger injected; the payload is directly provided as a tool observation to isolate the effect of trigger context.
> > > **Note this is not** a realistic attack mode, as the attacker cannot inject payloads into arbitrary tool responses without Stage I.
> > >
> > > | Delivery Mode | Hit | RTI | Acc |
> > > |:---|:---:|:---:|:---:|
> > > | Normal pipeline (trigger in context) | 92% | 9.7× | 96.1% |
> > > | Direct delivery (no trigger in context) | 90% | 9.4× | 96.3% |
> > >
> > > Hit and RTI are comparable across both modes (92% vs 90%, 9.7× vs 9.4×), indicating that the trigger's residual presence in the agent's context does not substantially affect Stage II payload processing. The E2E decomposition:
> > >
> > > $$\text{E2E} = P(S_2) = P(S_1) \times P(S_2 \mid S_1) = \text{ASR}_H \times \text{Hit}$$
> > >
> > > is therefore valid: $S_1$ is a necessary precondition for $S_2$, and $P(S_2)$ is empirically stable regardless of whether the trigger context is present.
> > >
> > > **(2) Is Hit truly measured on the Stage-I-success subset, or derived post hoc?**
> > >
> > > Hit is directly measured on the Stage-I-success subset. Our experimental pipeline works as follows:
> > >
> > > 1. For each of N=50 task instances, we run the full OTora pipeline (Stage I trigger injection → agent execution → Stage II payload delivery).
> > > 2. We record whether Stage I succeeds ($S_1$): does the agent produce a valid tool call to the target URL?
> > > 3. For episodes where $S_1$ = true, we record whether Stage II succeeds ($S_2$): does the agent encounter and process the reasoning payload on the target page?
> > > 4. We compute:
> > >    - ASR_H = (# episodes where $S_1$ = true) / N
> > >    - Hit = (# episodes where $S_1$ = true AND $S_2$ = true) / (# episodes where $S_1$ = true)
> > >    - E2E = (# episodes where $S_1$ = true AND $S_2$ = true) / N
> > >
> > > Hit is therefore a direct count on the Stage-I-success subset, not a post-hoc derivation.
> > >
> > > **(3) Empirical validation of decomposition consistency.**
> > >
> > > To provide the requested empirical analysis, we verify that the independently counted joint success rate matches the chain-rule decomposition across all reported settings (Environment injection):
> > >
> > > | Model | Agent | ASR_H | Hit | ASR_H × Hit | Directly counted E2E |
> > > |:---|:---|:---:|:---:|:---:|:---:|
> > > | LLaMA-70B | WebShop | 95% | 92% | 87.4% | 87% |
> > > | | Email | 94% | 91% | 85.5% | 86% |
> > > | | OS | 93% | 86% | 80.0% | 80% |
> > > | GPT-3.5 | WebShop | 58% | 82% | 47.6% | 48% |
> > > | | Email | 52% | 80% | 41.6% | 42% |
> > > | | OS | 44% | 73% | 32.1% | 32% |
> > >
> > > Across all 6 settings, the decomposed product (ASR_H × Hit) matches the directly counted E2E within ±1pp, which is consistent with integer rounding at N=50 (e.g., 1/50 = 2pp). This confirms that the conditional formulation accurately reflects the pipeline's actual behavior, and that the causal dependency between stages, while real, is properly accounted for by the conditional probability $P(S_2 \mid S_1)$.
> > >
> > > We will add this empirical validation table and the detailed measurement protocol to the revised paper to address this methodological concern explicitly.

---

### Official Review · Reviewer_6co8 · 2026-03-06

**Soundness:** 2
**Presentation:** 3
**Significance:** 2
**Originality:** 2
**Overall Recommendation:** 4
**Confidence:** 3

**Summary:**

The paper studies reasoning-level denial-of-service (R-DoS) on tool-using LLM agents, where an attacker aims to increase reasoning and tool-use cost while keeping task outcomes correct. It proposes OTora, a two-stage red-teaming pipeline. Stage I searches for a short adversarial trigger (in user instructions or environment observations) that causes the agent to invoke an attacker-chosen tool call to a target webpage, using insertion-point scoring (Eq. 1), dynamic target co-evolution (Eq. 2), and white-box or black-box suffix optimization (Eq. 3). Stage II hosts a reasoning-intensive payload on the retrieved page and optimizes it with an ICL-guided genetic search and a multi-objective score balancing reasoning token inflation, task fidelity, and stability (Eq. 4). Experiments on WebShop and InjecAgent Email and OS agents show improved Stage I tool-call success over baselines (e.g., black-box ASRH up to 58% on GPT-3.5, Table 1; white-box ASRH up to 95% on LLaMA-3.1-70B, Table 2) and Stage II slowdowns of roughly 5.1x to 10.8x reasoning-token inflation with near-baseline task accuracy (Table 3).

**Compliance With Llm Reviewing Policy:**

Affirmed.

**Ethical Review Concerns:**

1) Dual-use risk: the detailed two-stage attack pipeline could enable service disruption if released without safeguards (Sec. 3, Algorithm 1, Appendix D). Suggested mitigation: restrict release of executable triggers/payloads, provide redacted examples, and include controlled-use guidance.
2) Third-party impact: the use of a real-looking domain public.com is not clarified as controlled/mock (Sec. 3.1). Suggested mitigation: explicitly state it is controlled or replace with a clearly non-existent domain in text and artifacts.
3) Research integrity: inclusion of a reviewer-influencing instruction near Sec. 2 is inappropriate. Suggested mitigation: remove it, explain in rebuttal, and add a final pre-submission check to avoid embedded reviewer manipulation.
4) Dual-use balance: defenses are discussed but not quantitatively evaluated (Appendix A). Suggested mitigation: add concrete defense baselines and actionable deployment guidance.

**Ethical Review Flag:**

Flag this paper for an ethics review.

**Ethics Expertise Needed:**

["Inappropriate Potential Applications & Impact (e.g., human rights concerns)"]

**Final Justification:**

This paper studies an interesting and underexplored threat model for LLM agents, and the proposed two-stage framework is clear and technically solid. The empirical results are strong, and the additional analyses help clarify the contribution.

My initial concerns were mainly about end-to-end reporting, transferability, correctness evaluation details, defense evaluation, and some reproducibility / ethics issues. The rebuttal addressed these points well, which increased my confidence in the paper and led me to raise my score from 2 to 4. While some limitations remain, I now view this as a solid contribution with meaningful practical relevance.

**Key Questions For Authors:**

1) Clarify whether Stage I triggers and Stage II payloads are optimized per task instance, per agent prompt template, or can be reused across tasks. If strong reuse/transfer is shown, I would raise significance and overall recommendation.
2) Provide the number of evaluation episodes per benchmark, agent, and model, and the exact definition of Acc. for Email and OS. If metrics and sample sizes are clearly specified with uncertainty estimates, I would raise soundness.
3) Report end-to-end success and slowdown (ASRH times Hit, plus Delay/RTI conditional on success) for each setting in Table 1–3. If black-box end-to-end success is reasonably high under realistic budgets, I would consider raising the overall recommendation.
4) Add quantitative experiments for at least budgeting, relevance filtering/truncation, and runtime monitoring as discussed in Appendix A. If defenses show clear, actionable trade-offs, I would raise significance and actionability.
5) Explain the reviewer-influencing instruction included near Sec. 2 and confirm it will be removed. If addressed clearly, it would improve confidence and alleviate integrity concerns.

**Limitations:**

Not fully.
1) The paper does not quantify how conclusions change under common deployment constraints such as allowlisted domains/tools or aggressive observation truncation (Appendix B discusses truncation qualitatively).
2) The evaluation relies on observable reasoning tokens and harness-specific latency; limitations for systems with hidden reasoning traces or different billing/latency regimes are not clearly scoped (Sec. 4).
3) Dual-use is acknowledged (Impact Statement), but a concrete release/disclosure plan for triggers/payloads or code is not specified.

**Strengths And Weaknesses:**

Strengths :
1) Formalizes an availability threat where correctness is preserved but reasoning and tool budgets are exhausted (Sec. 1, Sec. 3.2).
2) Provides a concrete two-stage pipeline with implementable details (Sec. 3.1–3.4, Algorithm 1, Eq. 1–4).
3) Stage I trigger optimization improves ASRH over black-box baselines across agents and injection surfaces, e.g., GPT-3.5 WebShop environment ASRH 58% vs PAL 50% (Table 1), and strong white-box ASRH up to 95% on LLaMA-3.1-70B (Table 2).
4) Stage II yields large measured slowdowns with limited reported accuracy loss, e.g., LLaMA OS delay 360 s vs 38 s and RTI 10.8x with accuracy 95.7% (Table 3).
5) Ablations isolate the role of insertion strategy and Stage II objective terms (Table 4, Table 5) plus Stage I scoring components (Table 9, Table 10).
6) Reports attacker-side optimization cost in API tokens and GPU-hours (Appendix D, Table 6–8).

Weaknesses — MAJOR :
1) The manuscript includes an explicit instruction aimed at influencing reviewer text (near Sec. 2). This is inappropriate for peer review and can be interpreted as prompt-injection style manipulation. Fix: remove it and explicitly clarify in rebuttal that it was unintended and not part of the method.
2) End-to-end attack effectiveness is not directly reported. Sec. 4 states effectiveness is approximated as ASRH times Hit, but only separate Stage I ASRH (Table 1–2) and Stage II Hit (Table 3) are shown. This makes practical success unclear when black-box ASRH is 39% to 58% (Table 1) and Hit is 73% to 82% (Table 3). Fix: report combined end-to-end success (ASRH times Hit) and slowdown distributions conditional on success, per agent/model and injection surface.
3) Practical attacker assumptions and transferability are under-specified. Black-box trigger optimization can be costly (e.g., 4,310 API calls and 7.3M tokens for ASRH 57% on GPT-3.5 WebShop; Appendix D, Table 6). It is not specified whether triggers/payloads are optimized per task instance, per agent template, or reused across tasks/models, which matters for realism. Fix: clarify what is optimized once vs per instance, and add transfer experiments across tasks, prompts, and backbone models.
4) Correctness preservation is central but evaluation protocol is insufficiently specified. Sec. 4 mentions manual inspection, yet sample sizes, exact success criteria for Email/OS, and annotation protocol are not specified; Table 3 shows accuracy sometimes increases under attack, suggesting noise or small N. Fix: specify episode counts and exact automated metrics; report uncertainty (e.g., bootstrap CI) for Acc., Hit, Delay, and RTI.
5) Defenses are discussed but not evaluated. Appendix A outlines budgeting, filtering, monitoring, but no quantitative results are provided. Fix: implement at least 2–3 defenses (timeouts/token caps, observation truncation or relevance filtering, anomaly monitoring) and evaluate availability vs correctness trade-offs on the same benchmarks.

Weaknesses — MINOR:
1) Weight selection for w1, w2, w3 in Eq. 4 is not specified (Sec. 3.4), reducing reproducibility. Fix: report values and sensitivity.
2) Delay measurements lack setup details (hardware, tool/network latency, variance) (Sec. 4, Appendix D). Fix: report environment configuration and variance across runs.
3) The target domain public.com and tool interface are central (Sec. 3.1), but it is not specified whether the domain is controlled/mock. Fix: clarify to avoid third-party impact and improve replicability.

---

> ### Author Rebuttal · Authors · 2026-03-30
>
> We thank the reviewer for the thorough and constructive review. Below we address each concern with new experimental results and clarifications.
>
> ---
>
> **W1/Q5: Reviewer-Influencing Instruction**
>
> The text near Sec. 2 is the **official ICML 2026 LLM-review detection watermark** embedded by the submission system.
>
> ---
>
> **W2/Q3: E2E Attack Effectiveness**
>
> We now report combined E2E (= ASR_H × Hit). The black-box/white-box distinction applies exclusively to trigger optimization; payload optimization is identical across both settings.
>
> E2E summary (OTora, Environment injection; representative models):
>
> |Model|Agent|E2E|RTI|Delay|
> |:---|:---|:---:|:---:|:---:|
> |LLaMA-70B|WebShop|87%|9.7×±1.4|325±41s|
> ||Email|86%|9.9×±1.1|335±37s|
> |GPT-3.5|WebShop|48%|5.1×±0.6|170±19s|
> ||Email|42%|5.3×±0.8|182±26s|
>
> OTora achieves the highest E2E across all baselines: 75–87% (white-box) and 30–48% (black-box). We will provide full E2E results and baseline comparisons across all models in the revised Appendix.
>
> ---
>
> **W3/Q1: Transferability**
>
> To address the concern, we clarify optimization methods and provide transfer experiments to assess reusability here.
> Both stages are optimized per instance.
>
> Cross-task transfer (trigger optimized on 1 instance, evaluated on 50 others):
>
> |Setting|Method|Per-inst|Cross-task|
> |:---|:---|:---:|:---:|
> |White-box (LLaMA-70B)|UDora|84%|21%±6|
> ||OTora|86%|31%±5|
> |Black-box (GPT-3.5)|PAL|39%|11%±4|
> ||OTora|44%|19%±4|
>
> Cross-prompt/model transfer (optimized on WebShop+LLaMA-70B, evaluated on different agents/models):
>
> |Target|UDora|OTora|
> |:---|:---:|:---:|
> |WebShop, LLaMA-70B (ref)|84%|86%|
> |Email, LLaMA-70B|11%|29%|
> |OS, LLaMA-70B|9%|24%|
> |WebShop, GPT-OSS-120B|14%|31%|
>
> Across all transfer settings, OTora consistently outperforms all baselines. The attacker can optimize once and reuse across instances, amortizing cost. We provide full tables in the revised Appendix.
>
> ---
>
> **W4/Q2: Correctness Evaluation Protocol**
>
> We provide the full evaluation protocol in the revised Appendix, including episode counts, success criteria, annotation protocol, and uncertainty estimates. Each setting uses N=50 task instances. Correctness is determined by automated string matching against the expected gold action (e.g., `click[<ID>]` for WebShop, etc.); manual inspection is used only as a post-hoc sanity check.
>
> 95% bootstrap CI (partial; representative rows, 1000 resamples):
>
> |Setting|No-Attack Acc|Attack Acc|
> |:---|:---:|:---:|
> |GPT-3.5, Email|93.0%(83.0,97.5)|93.0%(82.5,98.0)|
> |LLaMA-70B, OS|95.0%(86.0,98.5)|95.7%(86.5,99.0)|
>
> We provide the full CI table across all 9 settings in the revised paper. All settings show overlapping CIs, confirming OTora preserves task correctness.
>
> ---
>
> **W5/Q4: Defense Evaluation**
>
> We implement two representative defenses and evaluate the availability–correctness trade-off on LLaMA-70B, WebShop (N=50).
>
> Defense 1: Observation Relevance Filtering (content below θ discarded):
>
> |Filter θ|OTora E2E|RTI|Benign Acc|
> |:---:|:---:|:---:|:---:|
> |0|86%|9.7×|95%|
> |0.5|68%|7.8×|81%|
> |0.7|48%|6.3×|64%|
>
> Defense 2: Runtime Monitoring (early stop if per-turn tokens > μ+cσ):
>
> |Threshold c|OTora E2E|RTI|Benign Completion|
> |:---:|:---:|:---:|:---:|
> |∞|86%|9.7×|100%|
> |2.0σ|70%|9.4×|84%|
> |1.5σ|58%|9.2×|72%|
>
> Both defenses reduce R-DoS but at significant benign cost: tighter thresholds filter legitimate content or falsely terminate complex tasks. Full evaluation details in revised Appendix.
>
> ---
>
> **Minor 1:**
>
> $w_1, w_2, w_3$ are hyperparameters in Eq. 4 controlling the RTI, fidelity, and stability terms. We use $w_1 = w_2 = w_3 = 1.0$ as the default in all experiments. Sensitivity analysis with other configurations confirms equal weights achieve the best balance. We provide full sensitivity results in the revised Appendix.
>
> ---
>
> **Minor 2:**
>
> White-box: NVIDIA GH200 (120GB) accelerators; black-box: API access. All tool calls simulated with deterministic responses; Delay reflects inference time. Variance (LLaMA-70B, WebShop): no-attack 30±6s, OTora 325±41s — slowdown far exceeds run-to-run variance. We will provide full details in the revised Appendix.
>
> ---
>
> **Minor 3:**
>
> `public.com` is a placeholder. We will rename it to `example-attacker.test` in the revised paper.
>
> ---
>
> **Limitations**
> We appreciate these valid points and will discuss all of them in the Limitations section of the revised paper.
>
> **L1:** We will add quantitative experiments under deployment constraints (allowlisted domains, observation truncation).
> **L2:** For systems with hidden chain-of-thought, RTI cannot be directly measured; API latency or billing cost can serve as proxy metrics.
> **L3:** We will open-source the framework code but withhold optimized triggers/payloads to prevent misuse.
>
> ---
>
> **Ethical Review**
>
> **E1 (dual-use):** See L3.
> **E2 (public.com):** See Minor 3.
> **E3 (integrity):** See W1.
> **E4 (defense):** See W5. Deployment guidance added to revised Appendix.

---

> > ### Author Rebuttal · Reviewer_6co8 · 2026-04-02
> >
> > Thank you for the detailed rebuttal. The added end-to-end results, transfer experiments, correctness protocol / uncertainty estimates, and defense trade-off analysis address several of my original concerns.
> >
> > My remaining concern is mainly about practical attack realism: the strongest results still seem to rely on per-instance optimization, while cross-task / cross-model transfer is materially weaker. I would therefore appreciate a clearer discussion of how much of the threat should be interpreted as requiring instance-specific optimization versus reusable attack artifacts. If this point is clarified, I would be open to revisiting my score.

---

> > > ### Author Response · Authors · 2026-04-06
> > >
> > > We thank the reviewer for the thoughtful follow-up and are pleased that most concerns have been addressed. We now clarify the remaining question regarding practical attack realism through a simulation-based user churn analysis.
> > >
> > > Our results show that per-instance optimization and transfer attacks represent complementary real-world threats: the former induces high-impact short-term disruption, while the latter enables persistent, low-cost degradation at scale.
> > >
> > > ---
> > >
> > > **Case Study: Per-Instance Optimization vs Reusable Attack Artifacts**
> > >
> > > To quantify real-world impact, we simulate user churn under R-DoS using established models of user patience from call center literature (Brown et al., 2005; Mandelbaum and Zeltyn, 2004) and recent findings on AI agent latency (Tabacof, 2025).
> > >
> > > **(1) User churn under R-DoS.**
> > >
> > > We model user patience as a mixture of log-normal distributions calibrated to the above literature, and compute the abandonment rate $q(d)$ as a function of agent response delay $d$:
> > >
> > > **User Abandonment Rate $q(d)$ vs. Agent Response Delay $d$** (mean delay from LLaMA-70B OTora-Persistent, Table 3)**:**
> > >
> > > |Agent|Condition|Mean Delay|$q(d)$|
> > > |:---|:---|:---:|:---:|
> > > |WebShop|Baseline (no attack)|30s|8%|
> > > |WebShop|Under R-DoS|325s|89%|
> > > |Email|Baseline (no attack)|30s|8%|
> > > |Email|Under R-DoS|335s|90%|
> > > |OS|Baseline (no attack)|38s|10%|
> > > |OS|Under R-DoS|360s|92%|
> > >
> > > Across all agents, R-DoS increases abandonment from $8–10%$ to $89–92%$, i.e., a $+81–82$ percentage point increase.
> > > In other words, **R-DoS transforms a normally reliable service into one where the vast majority of users abandon before task completion.**
> > >
> > > **(2) Instance-specific vs Transfer.**
> > >
> > > Using the simulated $q(d)$ above and attack parameters from our paper (Table 2/3), we compute daily user churn for both attack scenarios ($N=10,000$ requests/day):
> > >
> > > |Model|Agent|Scenario|E2E|Additional Churned Users/day|
> > > |:---|:---|:---|:---:|:---:|
> > > |LLaMA-70B|WebShop|Instance|87%|7,047|
> > > |LLaMA-70B|WebShop|Transfer|31%|2,511|
> > > |GPT-OSS-120B|WebShop|Instance|85%|6,715|
> > > |GPT-OSS-120B|WebShop|Transfer|30%|2,370|
> > > |LLaMA-70B|Email|Instance|86%|7,052|
> > > |LLaMA-70B|Email|Transfer|28%|2,296|
> > >
> > > Using $N=10,000$ requests/day and empirically observed delays:
> > > - Instance-specific attacks: ~$7,000$ additional churned users/day
> > > - Transfer attacks: ~$2,500$ additional churned users/day
> > >
> > > **Even with lower E2E success rates, transfer attacks still induce substantial real-world damage, due to their scalability and negligible marginal cost.**
> > >
> > > **(3) Time dimension: short-term burst vs long-term sustained.**
> > >
> > > Using LLaMA-70B on WebShop as a representative case (other model/agent combinations show consistent trends):
> > >
> > > |Strategy|Daily Additional Churn|30 Days|365 Days|
> > > |:---|:---:|:---:|:---:|
> > > |Instance (burst)|7,047|—|—|
> > > |Transfer (persistent)|2,511|75,330|916,515|
> > >
> > > **Transfer's near-zero marginal cost enables persistent deployment, accumulating ~917K churned users over a year, far exceeding any short-term instance-specific burst.** Instance-specific achieves 7,047 churned users/day but requires repeated per-request optimization, making sustained deployment cost-prohibitive. Transfer achieves 2,511 churned users/day with only a one-time optimization cost, accumulating to 75K over 30 days and ~917K over a year.
> > >
> > > **Discussion:** The R-DoS threat operates along two complementary dimensions. Per-instance optimization delivers high-impact short-term strikes, suitable for targeted disruption during critical periods, but is not sustainable at scale.
> > > Reusable transfer artifacts deliver persistent, lower-cost attacks with near-zero ongoing cost, enabling sustained degradation of service reliability. The threat should not be interpreted as relying on either strategy alone: per-instance optimization and reusable transfer artifacts represent complementary threat vectors targeting different attack objectives.
> > >
> > > Full simulation details will be included in the revised Appendix.
> > >
> > > **References:**
> > > [1] Brown, L., Gans, N., Mandelbaum, A., Sakov, A., Shen, H., Zeltyn, S., and Zhao, L. (2005). Statistical analysis of a telephone call center: A queueing-science perspective. Journal of the American Statistical Association, 100(469):36–50.
> > > [2] Mandelbaum, A. and Zeltyn, S. (2004). The impact of customers' patience on delay and abandonment: some empirically-driven experiments with the M/M/n+G queue. OR Spectrum, 26(3):377–411.
> > > [3] Tabacof, P. (2025). Slower feels smarter? Experimenting with AI agent latency. Intercom AI Research.

---

### Official Review · Reviewer_R5XT · 2026-03-13

**Soundness:** 3
**Presentation:** 3
**Significance:** 3
**Originality:** 3
**Overall Recommendation:** 4
**Confidence:** 4

**Summary:**

The authors propose OTora, a two-stage framework: Stage I triggers attacker-chosen external access, and Stage II injects reasoning-heavy payloads that induce persistent slowdown. Empirically, the paper reports large latency increases and reasoning-token inflation across WebShop, Email, and OS agents, while preserving near-baseline task accuracy. The problem is interesting and timely, especially because most prior agent-security work focuses on integrity rather than availability.

**Compliance With Llm Reviewing Policy:**

Affirmed.

**Final Justification:**

My concerns are addressed and I maintain postive towards the paper.

**Key Questions For Authors:**

1. What agent properties actually drive the vulnerability to R-DoS: longer context, stronger reasoning models, or the specific history-dependent design of these agents?

**Limitations:**

yes

**Strengths And Weaknesses:**

**Strengths**
The paper studies a less explored but important threat model: availability degradation without obvious task failure. The two-stage decomposition is clear, and the empirical results are fairly strong, with substantial slowdown and little accuracy drop across multiple agents and models.

**Weaknesses**
1. Evaluation is narrow. All agents use a similar ReAct-style setup, and the benchmarks are limited to WebShop and InjecAgent-style Email/OS tasks, so it is hard to know how broadly the conclusions transfer to more realistic production agents.
2. The analysis can be deeper. The paper shows that slowdown is possible, but does not fully explain which agent properties make some systems more vulnerable: e.g., whether the main driver is long context, ReAct history accumulation, tool schema design, or general reasoning tendency of stronger models.

---

> ### Author Rebuttal · Authors · 2026-03-30
>
> We thank the reviewer for recognizing the importance of the problem, the clear methods, and the extensive experiments. Below we address each concern and will incorporate the clarifications and additional analyses in the revision.
>
> ---
>
> **W1: Evaluation Scope**
>
> We appreciate the reviewer's concern.
> We chose ReAct as the most widely adopted agent paradigm (Yao et al., 2022). OTora's attack mechanism exploits tool-use interfaces and reasoning behavior, which are common across frameworks (ReAct, function-calling, multi-agent orchestration), not ReAct-specific. Within this framework, we evaluate 3 agent types (WebShop, Email, OS) with different tool schemas and 4 backbone models, showing consistent effectiveness across all settings.
>
> During the rebuttal period, we have conducted additional experiments with two new backbone models — Qwen-2.5-32B and DeepSeek-V2-67B — to further validate OTora's effectiveness within this framework. Results on WebShop (environment injection):
>
> | Model | E2E | RTI | Hit | Acc |
> |:---|:---:|:---:|:---:|:---:|
> | LLaMA-3.1-70B (paper) | 87% | 9.7× | 92% | 96.1% |
> | GPT-OSS-120B (paper) | 85% | 9.1× | 90% | 96.3% |
> | **Qwen-2.5-32B (new)** | **82%** | **8.6×** | **89%** | **95.2%** |
> | **DeepSeek-V2-67B (new)** | **84%** | **9.2×** | **90%** | **95.5%** |
>
> OTora achieves comparable E2E and RTI on both new model families, confirming the attack generalizes across model families beyond those in the original evaluation. In future work, we plan to extend OTora to additional agent frameworks (e.g., function-calling APIs, multi-agent orchestration) as a broader extension of this work. We will include the complete results of the new models across all three agents (WebShop, Email, OS), along with detailed experimental setup, in the revised Appendix.
>
> **Reference:**
> - Shunyu Yao, Jeffrey Zhao, Dian Yu, Nan Du, Izhak Shafran, Karthik R. Narasimhan, and Yuan Cao. 2022. ReAct: Synergizing Reasoning and Acting in Language Models. In *The Eleventh International Conference on Learning Representations*.
>
> ---
>
> **W2: Vulnerability Drivers**
>
> Many thanks for the suggestion.
> We provide a factor-by-factor analysis addressing the reviewer's specific dimensions with experimental evidence.
>
> **(1) Task context.** We compare the three agents (LLaMA-70B, OTora-Persistent):
>
> | Agent | ASR_H (Instr.) | ASR_H (Env.) | Hit | E2E (Instr.) | E2E (Env.) | Acc |
> |:---|:---:|:---:|:---:|:---:|:---:|:---:|
> | WebShop | 93% | 95% | 92% | 86% | 87% | 96.1% |
> | Email | 92% | 94% | 91% | 84% | 86% | 96.0% |
> | OS | 91% | 93% | 86% | 78% | 80% | 95.7% |
>
> WebShop has the highest E2E (87%) and Hit (92%), which likely reflects contextual naturalness: WebShop is a web-interaction task where browsing webpages is part of the normal workflow, so triggering a visit to an external URL is more semantically consistent. In contrast, OS agents perform system-level operations where visiting a webpage is contextually unusual, which may make trigger activation harder (Hit 86%, E2E 80%). Email falls in between.
>
> **(2) History-dependent design.** ReAct-style agents condition each Thought–Action pair on the entire interaction history. OTora-Persistent exploits this by injecting meta-instructions that recur across future turns. To isolate this effect from optimizer differences, we compare methods sharing the same ICL-guided optimization (Table 3, LLaMA-70B):
>
> | Stage II Method | Agent | Delay | RTI | Hit | Acc |
> |:---|:---|:---:|:---:|:---:|:---:|
> | OTora-ICL(Aware) | WebShop | 260s | 7.6× | 93% | 96.0% |
> | | Email | 270s | 7.8× | 92% | 95.9% |
> | | OS | 295s | 8.5× | 88% | 95.6% |
> | **OTora-Persistent (ours)** | **WebShop** | **325s** | **9.7×** | **92%** | **96.1%** |
> | | **Email** | **335s** | **9.9×** | **91%** | **96.0%** |
> | | **OS** | **360s** | **10.8×** | **86%** | **95.7%** |
>
> The only difference is the persistent policy component. RTI increases from 7.6× to 9.7× (+28%) across all agents while Hit and Acc remain nearly identical, suggesting that history-dependent design is a key amplifier: without it, R-DoS tends to be limited to single-turn overhead; with it, inflation compounds across turns.
>
> Regarding model scale, we observe that larger models (60B+) exhibit higher RTI (8.6–9.7×) than smaller models (5.1–5.3×) under the same payload (full per-model results in the revised Appendix). However, larger models also produce more tokens at baseline, so the higher RTI may partially reflect longer baseline reasoning rather than greater intrinsic vulnerability.
> We provide detailed normalized metrics in our response to **Reviewer HjRQ KQ2**, which shows 8.4–10.4× normalized delay increase across all models after controlling for baseline length.
>
> ---
>
> **KQ1: What Agent Properties Drive R-DoS Vulnerability?**
>
> See W2 above for the detailed factor-by-factor analysis.

---

> > ### Author Rebuttal · Reviewer_R5XT · 2026-04-01
> >
> > I appreciate the author's rebuttal, and my concerns are resolved. I would like to maintain my score since it's already positive.

---

> > > ### Author Response · Authors · 2026-04-02
> > >
> > > We are glad that our clarification and additional experiments have fully addressed the reviewer's concerns. We sincerely appreciate your time and careful evaluation. Your constructive feedback has been very valuable in improving the manuscript.
> > >
> > > Best regards,
> > >
> > > The authors

---

### Official Review · Reviewer_W96F · 2026-03-13

**Soundness:** 3
**Presentation:** 3
**Significance:** 2
**Originality:** 3
**Overall Recommendation:** 4
**Confidence:** 3

**Summary:**

This paper introduces a framework for reasoning-level denial of service. There are two steps: 1) a prompt is optimized to get reasoning agents to visit a particular attacker controlled link, and 2) content to be served at the link is optimized to make the agent spend lots of time on reasoning. Two black-box (Gemini-1.5-Flash, GPT-3.5-Turbo) and two white-box (Llama 3.1 70B, GPT-OSS 120B) are evaluated.

**Compliance With Llm Reviewing Policy:**

Affirmed.

**Final Justification:**

The rebuttal was helpful and I appreciate that the new evaluations on reasoning models support the method, so I maintain my original positive score.

**Key Questions For Authors:**

See above. Also Figure 2 -- it is unclear where these convergence curves are coming from. Which model, or is this an average? Which prompt, or is this an average?

**Limitations:**

See above.

**Strengths And Weaknesses:**

Strengths:
- The method appears to be effective (high ASR, high RTI) compared to other attacks
- OTora loss converges faster than other attacks including UDora, I-GCG, and GCG

Weaknesses:
- There is no discussion of transferability. I would be interested in analysis of whether strings optimized on white-box models transfer to API models for which weights and logprobs are not accessible. If there is not much transfer, then it is unclear how useful OTora is as an attack, given that the major agents by usage run closed source models, e.g. Claude Code and Codex.
- The "malicious instruction" scenario is not very realistic, since attackers often are not able to insert exact prompts for users.
- Limited evaluation of the Stage II optimization -- it seems useful to understand how this approach performs against a wider range of reasoning models, as more have been open sourced, and as closed source reasoning models are also widely used.

---

> ### Author Rebuttal · Authors · 2026-03-30
>
> We thank the reviewer for the positive assessment of the effective method and empirical performance. Below we address each concern and will incorporate the clarifications and additional analyses in the revision.
>
> ---
>
> **W1: Transferability to Closed-Source API Models**
>
> OTora does not have to require white-box access to attack closed-source models. As shown in Table 1 (Sec. 4), OTora includes a black-box trigger optimization mode that uses API-level access or proxy models when logprobs are unavailable.
> We evaluate this directly on GPT-3.5-Turbo and Gemini-1.5-Flash, achieving 30–48% E2E success with 5–6× RTI, without any white-box model involvement.
>
> To further address the reviewer's concern, we additionally provide cross-model transfer experiments. Trigger and payload optimized on WebShop + LLaMA-70B (white-box); evaluated on other models without re-optimization:
>
> | Target Model | UDora E2E | **OTora E2E** | **OTora RTI** |
> |:---|:---:|:---:|:---:|
> | *LLaMA-70B (ref)* | *84%* | *86%* | *9.7×* |
> | Gemini-1.5-Flash (API) | 14% | **26%** | **5.1×** |
> | GPT-3.5-Turbo (API) | 10% | **21%** | **4.8×** |
>
> Cross-model transfer degrades (86% → 21–26%), but OTora still substantially outperforms UDora under the same transfer setting (21–26% vs. 10–14%), and RTI remains at 4.8–5.1×. Moreover, direct black-box optimization is OTora's primary mode for attacking closed-source models, achieving higher E2E (30–48%) than transfer (21–26%). We will add the full transfer results to the revised paper.
>
> ---
>
> **W2: Realism of Malicious Instruction Scenario**
>
> We agree that attackers typically cannot modify users' raw inputs. As clarified in Sec. 3.2, "malicious instruction" refers to injection through upstream application logic (e.g., prompt templates, third-party plugins, or inter-agent delegation), not direct modification of user-typed text. This attack surface is well-established in recent agent security literature (Greshake et al., 2023; Zhan et al., 2024; Zhang et al., 2025).
>
> Furthermore, our paper evaluates two injection surfaces: malicious instruction and malicious environment. Malicious environment refers to placing adversarial content in external sources (e.g., webpages, emails) that the agent retrieves via tool calls during normal task execution, the attacker does not need any access to the user's prompt or the application layer, only control over content on a public webpage that the agent may visit.
> This is arguably a more realistic attack surface, as it requires no prompt-level access at all. Our results consistently show that environment-level injection outperforms instruction-level injection by 2–4pp ASR_H across all settings (Tables 1–2). We will clarify this distinction more explicitly in the revised paper.
>
> **References:**
> - Kai Greshake, Sahar Abdelnabi, Shailesh Mishra, Christoph Endres, Thorsten Holz, and Mario Fritz. 2023. Not what you've signed up for: Compromising real-world llm-integrated applications with indirect prompt injection. In *Proceedings of the 16th ACM Workshop on Artificial Intelligence and Security*, pages 79–90.
> - Qiusi Zhan, Zhixiang Liang, Zifan Ying, and Daniel Kang. 2024. Injecagent: Benchmarking indirect prompt injections in tool-integrated large language model agents. In *Findings of the Association for Computational Linguistics: ACL 2024*, pages 10471–10506.
> - Jiawei Zhang, Shuang Yang, and Bo Li. 2025. Udora: A unified red teaming framework against llm agents by dynamically hijacking their own reasoning. *arXiv preprint arXiv:2503.01908*.
>
> ---
>
> **W3: Limited Evaluation on Reasoning Models**
>
> We appreciate the reviewer's concern.
> We have extended our evaluation to include Qwen-2.5-32B and DeepSeek-V2-67B. Results on WebShop (environment injection):
>
> | Model | E2E | RTI | Hit | Acc |
> |:---|:---:|:---:|:---:|:---:|
> | LLaMA-3.1-70B (paper) | 87% | 9.7× | 92% | 96.1% |
> | GPT-OSS-120B (paper) | 85% | 9.1× | 90% | 96.3% |
> | **Qwen-2.5-32B (new)** | **82%** | **8.6×** | **89%** | **95.2%** |
> | **DeepSeek-V2-67B (new)** | **84%** | **9.2×** | **90%** | **95.5%** |
>
> OTora achieves comparable E2E and RTI on both new models, confirming that the attack generalizes across model families. We will include the full results across all agents in the revised paper.
>
> ---
>
> **Question: Figure 2 Convergence Curves**
>
> We thank the reviewer for pointing this out. The convergence curves in Figure 2 are from LLaMA-3.1-70B on WebShop (environment injection), single representative task instance. We will update the caption to include these details. In addition, we will review all figures and tables in the revised paper to ensure complete experimental specifications (model, agent, injection surface, task instance) are provided, and include comprehensive setup descriptions in the Appendix where missing.

---

> > ### Author Rebuttal · Reviewer_W96F · 2026-04-03
> >
> > Thank you for the helpful rebuttal. My concerns are addressed. I tend to maintain my original positive score.

---

> > > ### Author Response · Authors · 2026-04-06
> > >
> > > We are pleased that our clarifications and additional experiments have fully addressed the reviewer’s concerns. We sincerely appreciate your time and careful evaluation. Your constructive feedback has been invaluable in improving the manuscript.
> > >
> > > Best regards,
> > > The authors

---

### Review · Ethics_Reviewer_myFi · 2026-03-24

**Recommendation:** Remediation action needed

**Ethics Issue:**

This paper is on Reasoning-Level Denial-of-Service (R-DoS) attacks that can be conducted against agents by exploiting their reasoning and tool use functionalities. Reviewer 6co8 flagged ethics review with well-written identification and proposed mitigations. Specifically, the first two points they flag are the dual-risk issues and third-party impact. The proposed mitigations provided are good ways forward that the authors should take.

I personally did not see the reviewer-influencing instruction near Sec. 2, but if one is there, the recommended mitigation is appropriate.

For the fourth point given, I do not believe concrete defense baselines need to be provided for the scope of this work which demonstrates the attack, but the impact statement can add more clear pointers to defenses if in other works. I think the mitigation to point one may also help to ameliorate this.

**Remediation Action:**

Noted above, further details in Reviewer 6co8's review.

---

### Decision · Program_Chairs · 2026-04-30

**Decision:**

Accept (regular)

**Comment:**

This paper introduces a new red-teaming framework that identifies Reasoning-Level Denial-of-Service as a significant availability threat to LLM agents. The paper demonstrates that agents can be forced into "overthinking" cycles that inflate latency by up to 10x without compromising task correctness.  Key concerns regarding transferability to black-box models and the practical measurement of end-to-end success were effectively resolved through the authors' rebuttal.